# LEARNING ReLU NETWORKS TO HIGH UNIFORM ACCURACY IS INTRACTABLE

**Julius Berner**[1,*], **Philipp Grohs**[1,2,3,*], and **Felix Voigtlaender**[4,*]

[1]Faculty of Mathematics, University of Vienna, Austria
[2]Research Network Data Science @ Uni Vienna, University of Vienna, Austria
[3]RICAM, Austrian Academy of Sciences, Austria
[4]Mathematical Institute for Machine Learning and Data Science (MIDS), Catholic University of Eichstätt-Ingolstadt, Germany
[*]All authors contributed equally

## ABSTRACT

Statistical learning theory provides bounds on the necessary number of training samples needed to reach a prescribed accuracy in a learning problem formulated over a given target class. This accuracy is typically measured in terms of a generalization error, that is, an expected value of a given loss function. However, for several applications — for example in a security-critical context or for problems in the computational sciences — accuracy in this sense is not sufficient. In such cases, one would like to have guarantees for high accuracy on every input value, that is, with respect to the uniform norm. In this paper we precisely quantify the number of training samples needed for any conceivable training algorithm to guarantee a given uniform accuracy on any learning problem formulated over target classes containing (or consisting of) ReLU neural networks of a prescribed architecture. We prove that, under very general assumptions, the minimal number of training samples for this task scales exponentially both in the depth and the input dimension of the network architecture.

## 1 INTRODUCTION

The basic goal of supervised learning is to determine a function[1] $u : [0,1]^d \to \mathbb{R}$ from (possibly noisy) samples $(u(x_1), \ldots, u(x_m))$. As the function $u$ can take arbitrary values between these samples, this problem is, of course, not solvable without any further information on $u$. In practice, one typically leverages domain knowledge to estimate the structure and regularity of $u$ a priori, for instance, in terms of symmetries, smoothness, or compositionality. Such additional information can be encoded via a suitable *target class* $U \subset C([0,1]^d)$ that $u$ is known to be a member of. We are interested in the optimal accuracy for reconstructing $u$ that can be achieved by any algorithm which utilizes $m$ point samples. To make this mathematically precise, we assume that this accuracy is measured by a norm $\|\cdot\|_Y$ of a suitable Banach space $Y \supset U$. Formally, an algorithm can thus be described by a map $A : U \to Y$ that can query the function $u$ at $m$ points $x_i$ and that outputs a function $A(u)$ with $A(u) \approx u$ (see Section 2.1 for a precise definition that incorporates adaptivity and stochasticity). We will be interested in *upper and lower bounds* on the accuracy that can be reached by any such algorithm — equivalently, we are interested in the minimal number $m$ of point samples needed for any algorithm to achieve a given accuracy $\varepsilon$ for every $u \in U$. This $m$ would then establish a fundamental benchmark on the sample complexity (and the algorithmic complexity) of learning functions in $U$ to a given accuracy.

The choice of the Banach space $Y$ — in other words *how we measure accuracy* — is very crucial here. For example, statistical learning theory provides upper bounds on the optimal accuracy in terms of an expected loss, i.e., with respect to $Y = L^2([0,1]^d, d\mathbb{P})$, where $\mathbb{P}$ is a (generally unknown)

---

[1]In what follows, the input domain $[0,1]^d$ could be replaced by more general domains (for example Lipschitz domains) without any change in the later results. The unit cube $[0,1]^d$ is merely chosen for concreteness.

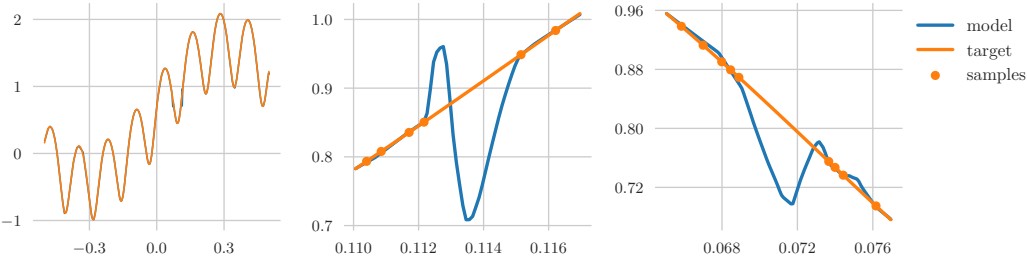

Figure 1: Even though the training of neural networks from data samples may achieve a small error *on average*, there are typically regions in the input space where the pointwise error is large. The target function in this plot is given by $x \mapsto \log(\sin(50x) + 2) + \sin(5x)$ (based on Adcock & Dexter, 2021) and the model is a feed-forward neural network. It is trained on $m = 1000$ uniformly distributed samples according to the hyperparameters in Tables 1 and 2 and achieves final $L^1$ and $L^\infty$ errors of $2.8 \cdot 10^{-3}$ and 0.19, respectively. The middle and right plots are zoomed versions of the left plot.

data generating distribution (Devroye et al., 2013; Shalev-Shwartz & Ben-David, 2014; Mohri et al., 2018; Kim et al., 2021). This offers a powerful approach to ensure a small *average* reconstruction error. However, there are many important scenarios where such bounds on the accuracy are not sufficient and one would like to obtain an approximation of $u$ that is close to $u$ not only on average, but that can be guaranteed to be close *for every $x \in [0,1]^d$*. This includes several applications in the sciences, for example in the context of the numerical solution of partial differential equations (Raissi et al., 2019; Han et al., 2018; Richter & Berner, 2022), any security-critical application, for example, facial ID authentication schemes (Guo & Zhang, 2019), as well as any application with a *distribution-shift*, i.e., where the data generating distribution is different from the distribution in which the accuracy is measured (Quiñonero-Candela et al., 2008). Such applications can only be efficiently solved if there exists an efficient algorithm $A$ that achieves *uniform accuracy*, i.e., a small error $\sup_{u \in U} \|u - A(u)\|_{L^\infty([0,1]^d)}$ with respect to the uniform norm given by $Y = L^\infty([0,1]^d)$, i.e., $\|f\|_{L^\infty([0,1]^d)} := \operatorname{esssup}_{x \in [0,1]^d} |f(x)|$.

Inspired by recent successes of *deep learning* across a plethora of tasks in machine learning (LeCun et al., 2015) and also increasingly the sciences (Jumper et al., 2021; Pfau et al., 2020), we will be particularly interested in the case where the target class $U$ consists of — or contains — realizations of (feed-forward) neural networks of a specific architecture[2]. Neural networks have been proven and observed to be extremely powerful in terms of their expressivity, that is, their ability to accurately approximate large classes of complicated functions with only relatively few parameters (Elbrächter et al., 2021; Berner et al., 2022). However, it has also been repeatedly observed that the *training* of neural networks (e.g., fitting a neural network to data samples) to high *uniform* accuracy presents a big challenge: conventional training algorithms (such as SGD and its variants) often find neural networks that perform well on average (meaning that they achieve a small generalization error), but there are typically some regions in the input space where the error is large (Fiedler et al., 2023); see Figure 1 for an illustrative example. This phenomenon has been systematically studied on an empirical level by Adcock & Dexter (2021). It is also at the heart of several observed instabilities in the training of deep neural networks, including *adversarial examples* (Szegedy et al., 2013; Goodfellow et al., 2015) or so-called *hallucinations* emerging in generative modeling, e.g., tomographic reconstructions (Bhadra et al., 2021) or machine translation (Müller et al., 2020).

Note that additional knowledge on the target functions could potentially help circumvent these issues, see Remark 1.3. However, for many applications, it is not possible to precisely describe the regularity of the target functions. We thus analyze the case where no additional information is given besides the fact that one aims to recover a (unknown) neural network of a specified architecture and regularization from given samples – i.e., we assume that $U$ contains a class of neural networks of a given architecture, subject to various regularization methods. This is satisfied in several applications of interest, e.g., *model extraction attacks* (Tramèr et al., 2016; He et al., 2022) and *teacher-student* settings (Mirzadeh et al., 2020; Xie et al., 2020). It is also in line with standard settings in the *statistical query literature*,

---

[2]By *architecture* we mean the number of layers $L$, as well as the number of neurons in each layer.

in *neural network identification*, and in statistical learning theory (Anthony & Bartlett, 1999; Mohri et al., 2018), see Section 1.1.

For such settings we can rigorously show that learning a class of neural networks is prone to instabilities. Specifically, any conceivable learning algorithm (in particular, any version of SGD), which recovers the neural network to high uniform accuracy, needs intractably many samples.

**Theorem 1.1.** *Suppose that $U$ contains all neural networks with $d$-dimensional input, ReLU activation function, $L$ layers of width up to $3d$, and coefficients bounded by $c$ in the $\ell^q$ norm. Assume that there exists an algorithm that reconstructs all functions in $U$ to uniform accuracy $\varepsilon$ from $m$ point samples. Then, we have*

$$m \geqslant \left(\frac{\Omega}{32d}\right)^d \cdot \varepsilon^{-d}, \quad \text{where} \quad \Omega := \begin{cases} \frac{1}{8 \cdot 3^{2/q}} \cdot c^L \cdot d^{1-\frac{2}{q}} & \text{if } q \leqslant 2 \\ \frac{1}{48} \cdot c^L \cdot (3d)^{(L-1)(1-\frac{2}{q})} & \text{if } q \geqslant 2. \end{cases}$$

Theorem 1.1 is a special case of Theorem 2.2 (covering $Y = L^p([0,1]^d)$ for all $p \in [1,\infty]$, as well as network architectures with arbitrary width) which will be stated and proven in Section 2.3.

To give a concrete example, we consider the problem of learning neural networks with ReLU activation function, $L$ layers of width at most $3d$, and coefficients bounded by $c$ to uniform accuracy $\varepsilon = 1/1024$. According to our results we would need at least

$$m \geqslant 2^d \cdot c^{dL} \cdot (3d)^{d(L-2)}$$

many samples — *the sample complexity thus depends exponentially on the input dimension $d$, the network width, and the network depth*, becoming intractable even for moderate values of $d, c, L$ (for $d = 15$, $c = 2$, and $L = 7$, the sample size $m$ would already have to exceed the estimated number of atoms in our universe). If, on the other hand, reconstruction only with respect to the $L^2$ norm were required, standard results in statistical learning theory (see, for example, Berner et al., 2020) show that $m$ only needs to depend polynomially on $d$. We conclude that uniform reconstruction is vastly harder than reconstruction with respect to the $L^2$ norm and, in particular, intractable. Our results are further corroborated by numerical experiments presented in Section 3 below.

**Remark 1.2.** *For other target classes $U$, uniform reconstruction is tractable (i.e., the number of required samples for recovery does not massively exceed the number of parameters defining the class). A simple example are univariate polynomials of degree less than $m$ which can be exactly determined from $m$ samples. One can show similar results for sparse multivariate polynomials using techniques from the field of compressed sensing (Rauhut, 2007). Further, one can show that approximation rates in suitable reproducing kernel Hilbert spaces with bounded kernel can be realized using point samples with respect to the uniform norm (Pozharska & Ullrich, 2022). Our results uncover an opposing behavior of neural network classes: There exist functions that can be arbitrarily well approximated (in fact, exactly represented) by small neural networks, but these representations cannot be inferred from samples. Our results are thus highly specific to classes of neural networks.*

**Remark 1.3.** *Our results do not rule out the possibility that there exist training algorithms for neural networks that achieve high accuracy on some restricted class of target functions, if the knowledge about the target class can be incorporated into the algorithm design. For example, if it were known that the target function can be efficiently approximated by polynomials one could first compute an approximating polynomial (using polynomial regression which is tractable) and then represent the approximating polynomial by a neural network. The resulting numerical problem would however be very different from the way deep learning is used in practice, since most neural network coefficients (namely those corresponding to the approximating polynomial) would be fixed a priori. Our results apply to the situation where such additional information on the target class $U$ is not available and no problem specific knowledge is incorporated into the algorithm design besides the network architecture and regularization procedure.*

We also complement the lower bounds of Theorem 1.1 with corresponding upper bounds.

**Theorem 1.4.** *Suppose that $U$ consists of all neural networks with $d$-dimensional input, ReLU activation function, $L$ layers of width at most $B$, and coefficients bounded by $c$ in the $\ell^q$ norm. Then, there exists an algorithm that reconstructs all functions in $U$ to uniform accuracy $\varepsilon$ from $m$ point samples with*

$$m \leqslant C^d \cdot \varepsilon^{-d}, \quad \text{where} \quad C := \begin{cases} \sqrt{d} \cdot c^L & \text{if } q \leqslant 2 \\ d^{1-\frac{1}{q}} \cdot c^L \cdot B^{(L-1)(1-\frac{2}{q})} & \text{if } q \geqslant 2. \end{cases}$$

Theorem 1.4 follows from Theorem 2.4 that will be stated in Section 2.4. We refer to Remark B.4 for a discussion of the gap between the upper and lower bounds.

**Remark 1.5.** *Our setting allows for an algorithm to choose the sample points* $(x_1, \ldots, x_m)$ *in an adaptive way for each* $u \in U$*; see Section 2.1 for a precise definition of the class of adaptive (possibly randomized) algorithms. This implies that even a very clever sampling strategy (as would be employed in active learning) cannot break the bounds established in this paper.*

**Remark 1.6.** *Our results also shed light on the impact of different regularization methods. While picking a stronger regularizer (e.g., a small value of q) yields quantitative improvements (in the sense of a smaller* $\Omega$*), the sample size* $m$ *required for approximation in* $L^\infty$ *can still increase exponentially with the input dimension* $d$*. However, this scaling is only visible for very small* $\varepsilon$*.*

## 1.1 RELATED WORK

Several other works have established "hardness" results for neural network training. For example, the seminal works by Blum & Rivest (1992); Vu (1998) show that for certain architectures the training process can be NP-complete. By contrast, our results do not directly consider algorithm runtime at all; our results are stronger in the sense of showing that even if it were possible to efficiently learn a neural network from samples, the necessary number of data points would be too large to be tractable.

We also want to mention a series of hardness results in the setting of *statistical query* (SQ) algorithms, see, e.g., Chen et al. (2022); Diakonikolas et al. (2020); Goel et al. (2020b); Reyzin (2020); Song et al. (2017). For instance, Chen et al. (2022) shows that any SQ algorithm capable of learning ReLU networks with two hidden layers and width $\text{poly}(d)$ up to $L^2$ error $1/\text{poly}(d)$ must use a number of samples that scales *superpolynomially* in $d$, or must use SQ queries with tolerance smaller than the reciprocal of any polynomial in $d$. In such SQ algorithms, the learner has access to an oracle that produces approximations (potentially corrupted by *adversarial noise*) of certain expectations $\mathbb{E}[h(X, u(X))]$, where $u$ is the unknown function to be learned, $X$ is a random variable representing the data, and $h$ is a function chosen by the learner (potentially subject to some restrictions, e.g. Lipschitz continuity). The possibility of the oracle to inject adversarial (instead of just stochastic) noise into the learning procedure — which does not entirely reflect the typical mathematical formulation of learning problems — is crucial for several of these results. We also mention that due to this possibility of adversarial noise, not every gradient-based optimization method (for instance, SGD) is strictly speaking an SQ algorithm; see also the works by Goel et al. (2020a, Page 3) and Abbe et al. (2021) for a more detailed discussion.

There also exist hardness results for learning algorithms based on *label queries* (i.e., noise-free point samples), which constitutes a setting similar to ours. More precisely, Chen et al. (2022) show that ReLU neural networks with constant depth and polynomial size constraints are not efficiently learnable up to a small squared loss with respect to a Gaussian distribution. However, the existing hardness results are in terms of *runtime* of the algorithm and are contingent on several (difficult and unproven) conjectures from the area of cryptography (the decisional Diffie-Hellmann assumption or the "Learning with Errors" assumption); the correctness of these conjectures in particular would imply that $P \neq NP$. By contrast, our results are completely free of such assumptions and show that the considered problem is *information-theoretically* hard, not just computationally.

As already hinted at in the introduction, our results further extend the broad literature on statistical learning theory (Anthony & Bartlett, 1999; Vapnik, 1999; Cucker & Smale, 2002b; Bousquet et al., 2003; Vapnik, 2013; Mohri et al., 2018). Specifically, we provide fully explicit upper and lower bounds on the *sample complexity* of (regularized) neural network *hypothesis classes*. In the context of PAC learning, we analyze the realizable case, where the target function is contained in the hypothesis class (Mohri et al., 2018, Theorem 3.20). Contrary to standard results, we do not pose any assumptions, such as IID, on the data distribution, and even allow for adaptive sampling. Moreover, we analyze the complexity for all $L^p$ norms with $p \in [1, \infty]$, whereas classical results mostly deal with the squared loss. As an example of such classical results, we mention that (bounded) hypothesis classes with finite pseudodimension $D$ can be learned to squared $L^2$ loss $\epsilon$ with $\mathcal{O}(D\epsilon^{-2})$ point samples; see e.g., Mohri et al. (2018, Theorem 11.8). Bounds for the pseudodimension of neural networks are readily available in the literature; see e.g., Bartlett et al. (2019). These bounds imply that learning ReLU networks in $L^2$ is tractable, in contrast to the $L^\infty$ setting.

Another related area is the identification of (equivalence classes of) neural network parameters from their input-output maps. While most works focus on scenarios where one has access to an infinite number of queries (Fefferman & Markel, 1993; Vlačić & Bölcskei, 2022), there are recent results employing only finitely many samples (Rolnick & Kording, 2020; Fiedler et al., 2023). Robust identification of the neural network parameters is sufficient to guarantee uniform accuracy, but it is not a necessary condition. Specifically, proximity of input-output maps does not necessarily imply proximity of corresponding neural network parameters (Berner et al., 2019). More generally, our results show that efficient identification from samples cannot be possible unless (as done in the previously mentioned works) further prior information is incorporated. In the same spirit, this restricts the applicability of model extraction attacks, such as *model inversion* or *evasion attacks* (Tramèr et al., 2016; He et al., 2022).

Our results are most closely related to recent results by Grohs & Voigtlaender (2021) where target classes consisting of neural network approximation spaces are considered. The results of Grohs & Voigtlaender (2021), however, are purely asymptotic. Since the asymptotic behavior incurred by the rate is often only visible for very fine accuracies, the results of Grohs & Voigtlaender (2021) cannot be applied to obtain concrete lower bounds on the required sample size. Our results are completely explicit in all parameters and readily yield practically relevant bounds. They also elucidate the role of adaptive sampling and different regularization methods.

## 1.2 NOTATION

For $d \in \mathbb{N}$, we denote by $C([0,1]^d)$ the space of continuous functions $f: [0,1]^d \to \mathbb{R}$. For a finite set $I$ and $(a_i)_{i \in I} \in \mathbb{R}^I$, we write $\sum_{i \in I} a_i := \frac{1}{|I|} \sum_{i \in I} a_i$. For $m \in \mathbb{N}$, we write $[m] := \{1, \ldots, m\}$. For $A \subset \mathbb{R}^d$, we denote by $A^\circ$ the set of interior points of $A$. For any subset $A$ of a vector space $V$, any $c \in \mathbb{R}$, and any $y \in V$, we further define $y + c \cdot A := \{y + ca: a \in A\}$. For a matrix $W \in \mathbb{R}^{n \times k}$ and $q \in [1, \infty)$, we write $\|W\|_{\ell^q} := \left( \sum_{i,j} |W_{i,j}|^q \right)^{1/q}$, and for $q = \infty$ we write $\|W\|_{\ell^\infty} := \max_{i,j} |W_{i,j}|$. For vectors $b \in \mathbb{R}^n$, we use the analogously defined notation $\|b\|_{\ell^q}$.

## 2 MAIN RESULTS

This section contains our main theoretical results. We introduce the considered classes of algorithms in Section 2.1 and target classes in Section 2.2. Our main lower and upper bounds are formulated and proven in Section 2.3 and Section 2.4, respectively.

### 2.1 ADAPTIVE (RANDOMIZED) ALGORITHMS BASED ON POINT SAMPLES

As described in the introduction, our goal is to analyze how well one can recover an unknown function $u$ from a target class $U$ in a Banach space $Y$ based on point samples. This is one of the main problems in *information-based complexity* (Traub, 2003), and in this section we briefly recall the most important related notions.

Given $U \subset C([0,1]^d) \cap Y$ for a Banach space $Y$, we say that a map $A: U \to Y$ is an *adaptive deterministic method using $m \in \mathbb{N}$ point samples* if there are $f_1 \in [0,1]^d$ and mappings

$$f_i: \left([0,1]^d\right)^{i-1} \times \mathbb{R}^{i-1} \to [0,1]^d, \quad i = 2, \ldots, m, \quad \text{and} \quad Q: \left([0,1]^d\right)^m \times \mathbb{R}^m \to Y$$

such that for every $u \in U$, using the point sequence $\mathbf{x}(u) = (x_1, \ldots, x_m) \subset [0,1]^d$ defined as

$$x_1 = f_1, \quad x_i = f_i(x_1, \ldots, x_{i-1}, u(x_1), \ldots, u(x_{i-1})), \quad i = 2, \ldots, m, \tag{1}$$

the map $A$ is of the form $A(u) = Q(x_1, \ldots, x_m, u(x_1), \ldots, u(x_m)) \in Y$.

The set of all deterministic methods using $m$ point samples is denoted by $\mathrm{Alg}_m(U, Y)$. In addition to such deterministic methods, we also study randomized methods defined as follows: A tuple $(\mathbf{A}, \mathbf{m})$ is called an *adaptive random method using $m \in \mathbb{N}$ point samples on average* if $\mathbf{A} = (A_\omega)_{\omega \in \Omega}$ where $(\Omega, \mathcal{F}, \mathbb{P})$ is a probability space, and where $\mathbf{m}: \Omega \to \mathbb{N}$ is such that the following conditions hold:

1. $\mathbf{m}$ is measurable, and $\mathbb{E}[\mathbf{m}] \leqslant m$;
2. $\forall u \in U: \omega \mapsto A_\omega(u)$ is measurable with respect to the Borel $\sigma$-algebra on $Y$;

3. $\forall \, \omega \in \Omega : \; A_\omega \in \mathrm{Alg}_{\mathbf{m}(\omega)}(U, Y)$.

The set of all random methods using $m$ point samples on average will be denoted by $\mathrm{Alg}_m^{MC}(U, Y)$, since such methods are sometimes called *Monte-Carlo* (MC) algorithms.

For a target class $U$, we define the *optimal (randomized) error* as

$$\mathrm{err}_m^{MC}(U, Y) := \inf_{(\mathbf{A}, \mathbf{m}) \in \mathrm{Alg}_m^{MC}(U, Y)} \; \sup_{u \in U} \; \mathbb{E}\left[ \| u - A_\omega(u) \|_Y \right]. \tag{2}$$

We note that $\mathrm{Alg}_m(U, Y) \subset \mathrm{Alg}_m^{MC}(U, Y)$, since each deterministic method can be interpreted as a randomized method over a trivial probability space.

## 2.2 Neural network classes

We will be concerned with target classes related to ReLU neural networks. These will be defined in the present subsection. Let $\varrho : \mathbb{R} \to \mathbb{R}$, $\varrho(x) = \max\{0, x\}$, be the *ReLU activation function*. Given a *depth* $L \in \mathbb{N}$, an *architecture* $(N_0, N_1, \dots, N_L) \in \mathbb{N}^{L+1}$, and *neural network coefficients*

$$\Phi = \left( (W^i, b^i) \right)_{i=1}^L \in \bigtimes_{i=1}^L \left( \mathbb{R}^{N_i \times N_{i-1}} \times \mathbb{R}^{N_i} \right),$$

we define their *realization* $R(\Phi) \in C(\mathbb{R}^{N_0}, \mathbb{R}^{N_L})$ as

$$R(\Phi) := \phi^L \circ \varrho \circ \phi^{L-1} \circ \cdots \circ \varrho \circ \phi^1$$

where $\varrho$ is applied componentwise and $\phi^i : \mathbb{R}^{N_{i-1}} \to \mathbb{R}^{N_i}$, $x \mapsto W^i x + b^i$, for $i \in [L]$. Given $c > 0$ and $q \in [1, \infty]$, define the class

$$\mathcal{H}_{(N_0, \dots, N_L), c}^q := \left\{ R(\Phi) : \; \Phi \in \bigtimes_{i=1}^L \left( \mathbb{R}^{N_i \times N_{i-1}} \times \mathbb{R}^{N_i} \right) \text{ and } \| \Phi \|_{\ell^q} \leqslant c \right\},$$

where $\| \Phi \|_{\ell^q} := \max_{1 \leqslant i \leqslant L} \max\{ \| W^i \|_{\ell^q}, \| b^i \|_{\ell^q} \}$.

To study target classes related to neural networks, the following definition will be useful.

**Definition 2.1.** *Let* $U, \mathcal{H} \subset C([0,1]^d)$. *We say that* $U$ *contains a copy of* $\mathcal{H}$, *attached to* $u_0 \in U$ *with constant* $c_0 \in (0, \infty)$, *if* $u_0 + c_0 \cdot \mathcal{H} \subset U$.

## 2.3 Lower bound

The following result constitutes the main result of the present paper. Theorem 1.1 readily follows from it as a special case.

**Theorem 2.2.** *Let* $L \in \mathbb{N}_{\geqslant 3}$, $d, B \in \mathbb{N}$, $p, q \in [1, \infty]$, *and* $c \in (0, \infty)$. *Suppose that the target class* $U \subset C([0,1]^d)$ *contains a copy of* $\mathcal{H}_{(d, B, \dots, B, 1), c}^q$ *with constant* $c_0 \in (0, \infty)$, *where the* $B$ *in* $(d, B, \dots, B, 1)$ *appears* $L - 1$ *times. Then, for any* $s \in \mathbb{N}$ *with* $s \leqslant \min\left\{ \frac{B}{3}, d \right\}$ *we have*

$$\mathrm{err}_m^{MC}(U, L^p([0,1]^d)) \geqslant c_0 \cdot \frac{\Omega}{(32s)^{1 + \frac{s}{p}}} \cdot m^{-\frac{1}{p} - \frac{1}{s}},$$

*where*

$$\Omega := \begin{cases} \frac{1}{8 \cdot 3^{2/q}} \cdot c^L \cdot s^{1 - \frac{2}{q}} & \text{if } q \leqslant 2 \\ \frac{1}{48} \cdot c^L \cdot B^{(L-1)(1 - \frac{2}{q})} & \text{if } q \geqslant 2. \end{cases}$$

*Proof.* This follows by combining Theorem A.5 with Lemmas A.2 and A.3 in the appendix. $\square$

**Remark 2.3.** *For* $p \ll \infty$, *the bound from above does not necessarily imply that an intractable number of training samples is needed. This is a reflection of the fact that efficient learning is possible (at least if one only considers the number of training samples and not the runtime of the algorithm) in this regime. Indeed, it is well-known in statistical learning theory that one obtains learning bounds based on the entropy numbers (w.r.t. the* $L^\infty$ *norm) of the class of target functions, when the error is measured in* $L^2$, *see, for instance, Cucker & Smale (2002a, Proposition 7). The* $\varepsilon$-*entropy numbers of a class of neural networks with* $L$ *layers and* $w$ *(bounded) weights scale linearly in* $w, L$ *and*

*logarithmically in $1/\varepsilon$, so that one gets tractable $L^2$ learning bounds. By interpolation for $L^p$ norms (noting that in our case the target functions are bounded, so that the $L^\infty$ reconstruction error is bounded, even though the decay with $m$ is very bad), this also implies $L^p$ learning bounds, but these get worse and worse as $p \to \infty$. We remark that these learning bounds are based on empirical risk minimization, which might be computationally infeasible (Vu, 1998); since our lower bounds should hold for any feasible algorithm (irrespective of its computational complexity), this means that one cannot expect to get an intractable lower bound for $p \ll \infty$ in our setting.*

The idea of the proof of Theorem 2.2 (here only presented for $u_0 = 0$ and $s = d$, which implies that $B \geqslant 3d$) is as follows:

1. We first show (see Lemmas A.2 and A.3) that the neural network set $\mathcal{H}^q_{(d,B,\ldots,B,1),c}$ contains a large class of "bump functions" of the form $\lambda \cdot \vartheta_{M,y}$. Here, $\vartheta_{M,y}$ is supported on the set $y + \left[-\frac{1}{M}, \frac{1}{M}\right]^d$ and satisfies $\|\vartheta_{M,y}\|_{L^p([0,1]^d)} \asymp M^{-d/p}$, where $M \in \mathbb{N}$ and $y \in [0,1]^d$ can be chosen arbitrarily; see Lemma A.1. The size of the scaling factor $\lambda = \lambda(M, c, q, d, L)$ depends crucially on the regularization parameters $c$ and $q$. This is the main technical part of the proof, requiring to construct suitable neural networks adhering to the imposed $\ell^q$ restrictions on the weights for which $\lambda$ is as big as possible.

2. If one learns using $m$ points samples $x_1, \ldots, x_m$ and if $M = \mathcal{O}(m^{1/d})$, then a volume packing argument shows that there exists $y \in [0,1]^d$ such that $\vartheta_{M,y}(x_i) = 0$ for all $i \in [m]$. This means that the learner cannot distinguish the function $\lambda \cdot \vartheta_{M,y} \in \mathcal{H}^q_{(d,B,\ldots,B,1),c}$ from the zero function and will thus make an error of roughly $\|\lambda \cdot \vartheta_{M,y}\|_{L^p} \asymp \lambda \cdot M^{-d/p}$. This already implies the lower bound in Theorem 2.2 for the case of *deterministic* algorithms.

3. To get the lower bound for *randomized* algorithms using $m$ point samples on average, we employ a technique from information-based complexity (see, e.g., Heinrich, 1994): We again set $M = \mathcal{O}(m^{1/d})$ and define $(y_\ell)_{\ell \in [M/2]^d}$ as the nodes of a uniform grid on $[0,1]^d$ with width $2/M$. Using a volume packing argument, we then show that for any choice of $m$ sampling points $x_1, \ldots, x_m$, "at least half of the functions $\vartheta_{M,y_\ell}$ avoid all the sampling points", i.e., for at least half of the indices $\ell$, it holds that $\vartheta_{M,y_\ell}(x_i) = 0$ for all $i \in [m]$. A learner using the samples $x_1, \ldots, x_m$ can thus not distinguish between the zero function and $\lambda \cdot \vartheta_{M,y_\ell} \in \mathcal{H}^q_{(d,B,\ldots,B,1),c}$ for at least half of the indices $\ell$. Therefore, any *deterministic* algorithm will make an error of $\Omega(\lambda \cdot M^{-d/p})$ *on average with respect to* $\ell$.

4. Since each randomized algorithm $\boldsymbol{A} = (A_\omega)_{\omega \in \Omega}$ is a collection of deterministic algorithms and since taking an average commutes with taking the expectation, this implies that any randomized algorithm will have an expected error of $\Omega(\lambda \cdot M^{-d/p})$ *on average with respect to* $\ell$. This easily implies the stated bound.

As mentioned in the introduction, we want to emphasize that well-trained neural networks can indeed exhibit such bump functions, see Figure 1 and Adcock & Dexter (2021); Fiedler et al. (2023).

## 2.4 UPPER BOUND

In this section we present our main upper bound, which directly implies the statement of Theorem 1.4.

**Theorem 2.4.** *Let $L, d \in \mathbb{N}$, $q \in [1, \infty]$, $c \in (0, \infty)$, and $N_1, \ldots, N_{L-1} \in \mathbb{N}$. Then, we have*

$$\mathrm{err}_m^{MC}\big(\mathcal{H}^q_{(d,N_1,\ldots,N_{L-1},1),c}, L^\infty([0,1]^d)\big) \leqslant \begin{cases} \sqrt{d} \cdot c^L \cdot m^{-\frac{1}{d}} & \text{if } q \leqslant 2 \\ \sqrt{d} \cdot c^L \cdot (\sqrt{d} \cdot N_1 \cdots N_{L-1})^{1-\frac{2}{q}} \cdot m^{-\frac{1}{d}} & \text{if } q \geqslant 2. \end{cases}$$

*Proof.* This follows by combining Lemmas B.2 and B.3 in the appendix. $\square$

Let us outline the main idea of the proof. We first show that each neural network $R(\Phi) \in \mathcal{H}^q_{(N_0,\ldots,N_L),c}$ is Lipschitz-continuous, where the Lipschitz constant can be conveniently bounded in terms of the parameters $N_0, \ldots, N_L, c, q$, see Lemma B.2 in the appendix. In Lemma B.3, we then show that any function with moderate Lipschitz constant can be reconstructed from samples by piecewise constant interpolation.

## 3 NUMERICAL EXPERIMENTS

Having established fundamental bounds on the performance of any learning algorithm, we want to numerically evaluate the performance of commonly used deep learning methods. To illustrate our main result in Theorem 2.2, we estimate the error in (2) by a tractable approximation in a student-teacher setting. Specifically, we estimate the minimal error over neural network target functions ("teachers") $\widehat{U} \subset \mathcal{H}^q_{(d,N_1,\ldots,N_{L-1},1),c}$ for deep learning algorithms $\widehat{A} \subset \mathrm{Alg}^{MC}_m(U, L^p)$ via Monte-Carlo sampling, i.e.,

$$\widehat{\mathrm{err}}_m\left(\widehat{U}, L^p; \widehat{A}\right) := \inf_{(\mathbf{A},\mathbf{m})\in\widehat{A}} \sup_{u\in\widehat{U}} {\textstyle\sum}_{\omega\in\widehat{\Omega}} \left({\textstyle\sum}_{j\in[J]} \left(u(X_j) - A_\omega(u)(X_j)\right)^p\right)^{1/p}, \quad (3)$$

where $(X_j)^J_{j=1}$ are independent evaluation samples uniformly distributed on[3] $[-0.5, 0.5]^d$ and $\widehat{\Omega}$ represents the seeds for the algorithms.

We obtain teacher networks $u \in \mathcal{H}^\infty_{(d,N_1,\ldots,N_{L-1},1),c}$ by sampling their coefficients $\Phi$ componentwise according to a uniform distribution on $[-c, c]$. For every algorithm $(\mathbf{A}, \mathbf{m}) \in \widehat{A}$ and seed $\omega \in \widehat{\Omega}$ we consider point sequences $\mathbf{x}(u)$ uniformly distributed in $[-0.5, 0.5]^d$ with $\mathbf{m}(\omega) = m$. The corresponding point samples are used to train the coefficients of a neural network ("student") using the Adam optimizer (Kingma & Ba, 2015) with exponentially decaying learning rate. We consider input dimensions $d = 1$ and $d = 3$, for each of which we compute the error in (3) for 4 different sample sizes $m$ over 40 teacher networks $u$. For each combination, we train student networks with 3 different seeds, 3 different widths, and 3 different batch-sizes. In summary, this yields $2 \cdot 4 \cdot 40 \cdot 3 \cdot 3 \cdot 3 = 8640$ experiments each executed on a single GPU. The precise hyperparameters can be found in Tables 1 and 3 in Appendix C.

Figure 2 shows that there is a clear gap between the errors $\widehat{\mathrm{err}}_m(\widehat{U}, L^p; \widehat{A})$ for $p \in \{1, 2\}$ and $p = \infty$. Especially in the one-dimensional case, the rate $\widehat{\mathrm{err}}_m(\widehat{U}, L^\infty; \widehat{A})$ w.r.t. the number of samples $m$ also seems to stagnate at a precision that might be insufficient for certain applications. Figure 3 illustrates that the errors are caused by spikes of the teacher network which are not covered by any sample. Note that this is very similar to the construction in the proof of our main result, see Section 2.3.

In general, the rates worsen when considering more teacher networks $\widehat{U}$ and improve when considering further deep learning algorithms $\widehat{A}$, including other architectures or more elaborate training and sampling schemes. Note, however, that each setting needs to be evaluated for a number of teacher networks, sample sizes, and seeds. We provide an extensible implementation[4] in PyTorch (Paszke et al., 2019) featuring multi-node experiment execution and hyperparameter tuning using Ray Tune (Liaw et al., 2018), experiment tracking using Weights & Biases and TensorBoard, and flexible experiment configuration. Building upon our work, research teams with sufficient computational resources can provide further numerical evidence on an even larger scale.

## 4 DISCUSSION AND LIMITATIONS

**Discussion.** We derived fundamental upper and lower bounds for the number of samples needed for any algorithm to reconstruct an arbitrary function from a target class containing realizations of neural networks with ReLU activation function of a given architecture and subject to $\ell^q$ regularization constraints on the network coefficients, see Theorems 2.2 and 2.4. These bounds are completely explicit in the network architecture, the type of regularization, and the norm in which the reconstruction error is measured. We observe that our lower bounds are severely more restrictive if the error is measured in the uniform $L^\infty$ norm rather than the (more commonly studied) $L^2$ norm. Particularly, learning a class of neural networks with ReLU activation function with moderately high accuracy in the $L^\infty$ norm is intractable for moderate input dimensions, as well as network widths and depths. We anticipate that further investigations into the sample complexity of neural network classes can eventually contribute to a better understanding of possible circumstances under which it is possible to design reliable deep learning algorithms and help explain well-known instability phenomena such

---

[3]To have centered input data, we consider the hypercube $[-0.5, 0.5]^d$ in our experiments. Note that this does not change any of the theoretical results.

[4]The code can be found at `https://github.com/juliusberner/theory2practice`.

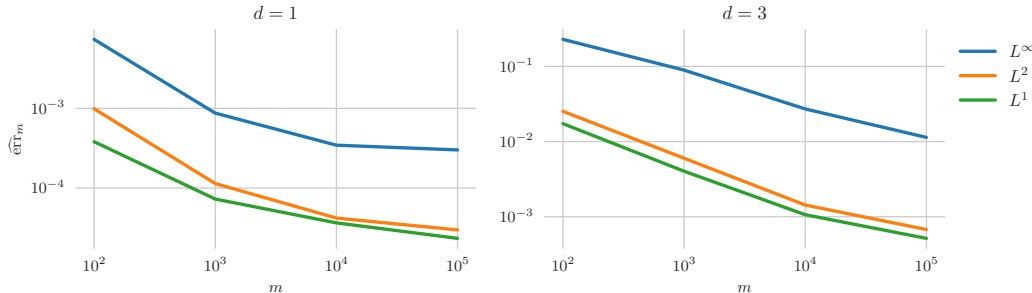

Figure 2: Evaluation of the error in (3) for $p \in \{1, 2, \infty\}$, input dimensions $d \in \{1, 3\}$, sample sizes $m \in \{10^2, 10^3, 10^4, 10^5\}$, and hyperparameters given in Tables 1 and 3.

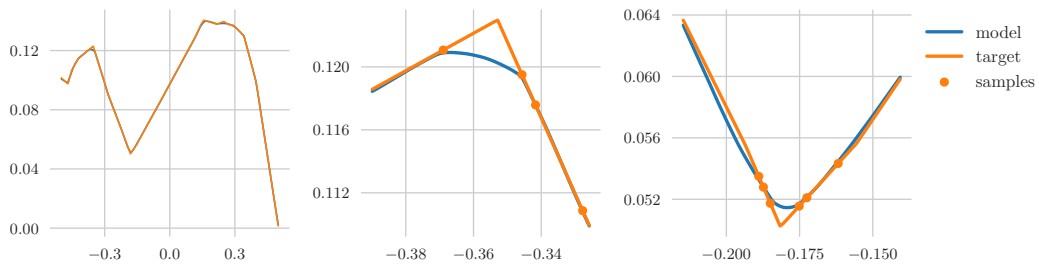

Figure 3: Target function ("teacher"), samples, and model of the deep learning algorithm ("student") attaining the min-max value in (3) for $m = 100$ and $p = \infty$ in the experiment depicted in Figure 2. The middle and right plots are zoomed versions of the left plot. The $L^\infty$ error ($2.7 \cdot 10^{-3}$) is about one magnitude larger than the $L^2$ and $L^1$ errors ($3.9 \cdot 10^{-4}$ and $2.4 \cdot 10^{-4}$), which is caused by spikes of the teacher network between samples.

as adversarial examples. Such an understanding can be beneficial in assessing the potential and limitations of machine learning methods applied to security- and safety-critical scenarios.

**Limitations and Outlook.** We finally discuss some possible implications and also limitations of our work. First of all, our results are highly specific to neural networks with the ReLU activation function. We expect that obtaining similar results for other activation functions will require substantially new methods. We plan to investigate this in future work.

The explicit nature of our results reveal a discrepancy between the lower and upper bound, especially for high dimensions. We conjecture that both the current upper and lower bounds are not quite optimal. Determining to which extent one can tighten the bounds is an interesting open problem.

Our analysis is a worst-case analysis in the sense that we show that for any given algorithm $A$, there exists at least one $u$ in our target class $U$ on which $A$ performs poorly. The question of whether this poor behavior is actually generic will be studied in future work. One way to establish such generic results could be to prove that our considered target classes contain copies of neural network realizations attached to many different $u$'s.

Finally, we consider target classes $U$ that contain all realizations of neural networks with a given architecture subject to different regularizations. This can be justified as follows: Whenever a deep learning method is employed to reconstruct a function $u$ by representing it approximately by a neural network (without further knowledge about $u$), a natural minimal requirement is that the method should perform well if the sought function is in fact equal to a neural network. However, if additional problem information about $u$ can be incorporated into the learning problem it may be possible to overcome the barriers shown in this work. The degree to which this is possible, as well as the extension of our results to other architectures, such as convolutional neural networks, transformers, and graph neural networks will be the subject of future work.

ACKNOWLEDGMENTS

The research of Julius Berner was supported by the Austrian Science Fund (FWF) under grant I3403-N32 and by the Vienna Science and Technology Fund (WWTF) under grant ICT19-041. The computational results presented have been achieved in part using the Vienna Scientific Cluster (VSC). Felix Voigtlaender acknowledges support by the DFG in the context of the Emmy Noether junior research group VO 2594/1-1.

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

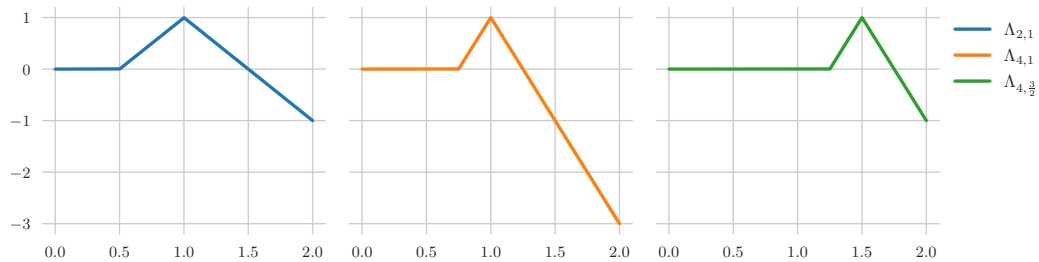

Figure 4: Plots of the function $\Lambda_{M,\sigma}$ in Equation (4) for $(M, \sigma) \in \{(2,1), (4,1), (4, \frac{3}{2})\}$.

## A   PROOF OF THE LOWER BOUND IN SECTION 2.3

### A.1   CONSTRUCTION OF HAT FUNCTIONS IMPLEMENTED BY RELU NETWORKS

For $d \in \mathbb{N}$, $M > 0$, $\sigma \in \mathbb{R}$, $s \in [d]$, and $y \in \mathbb{R}^d$, define

$$\Lambda_{M,\sigma}: \quad \mathbb{R} \to (-\infty, 1], \quad t \mapsto \begin{cases} 0 & \text{if } t \leqslant \sigma - \frac{1}{M} \\ 1 - M \cdot |t - \sigma| & \text{if } t \geqslant \sigma - \frac{1}{M}, \end{cases} \tag{4}$$

and furthermore

$$\Delta_{M,y}^{(s)}: \quad \mathbb{R}^d \to (-\infty, 1], \quad x \mapsto \left( \sum_{i=1}^{s} \Lambda_{M,y_i}(x_i) \right) - (s-1),$$

$$\vartheta_{M,y}^{(s)}: \quad \mathbb{R}^d \to [0,1], \quad x \mapsto \varrho(\Delta_{M,y}^{(s)}(x)),$$

where, as before, $\varrho : \mathbb{R} \to \mathbb{R}$, $x \mapsto \max\{0, x\}$, denotes the ReLU activation function. A plot of $\Lambda_{M,\sigma}$ is shown in Figure 4.

With these definitions, the function $\vartheta_{M,y}^{(s)}$ satisfies the following properties:

**Lemma A.1.** *For $d \in \mathbb{N}$, $s \in [d]$, $M \geqslant 1$, $y \in [0,1]^d$, and $p \in (0, \infty]$, we have*

$$\operatorname{supp} \vartheta_{M,y}^{(s)} \subset y + (M^{-1} \cdot [-1,1]^s \times \mathbb{R}^{d-s})$$

*and*

$$\frac{1}{2} \cdot (2s)^{-s/p} \cdot M^{-s/p} \leqslant \|\vartheta_{M,y}^{(s)}\|_{L^p([0,1]^d)} \leqslant 2^{s/p} \cdot M^{-s/p}.$$

*Proof.* Let us first give a quick overview of the proof. The statement on the support of $\vartheta_{M,y}^{(s)}$ follows by observing that $\Delta_{M,y}^{(s)}(x) > 0$ can only happen if $\Lambda_{M,y_i}(x_i) > 0$ for all $i \in [s]$. As $0 \leqslant \vartheta_{M,y}^{(s)} \leqslant 1$, the upper bound on the $L^p([0,1]^d)$ norm can then be estimated by the Lebesgue measure of the intersection of the support of $\vartheta_{M,y}^{(s)}$ and the hypercube $[0,1]^d$. For the lower bound we compute the measure of the intersection with a subset of the support on which it holds that $\vartheta_{M,y}^{(s)} \geqslant \frac{1}{2}$.

We start by proving the statement on the support of $\vartheta_{M,y}^{(s)}$. If $0 \neq \vartheta_{M,y}^{(s)}(x)$, then $\Delta_{M,y}^{(s)}(x) > 0$, meaning $\sum_{i=1}^{s} \Lambda_{M,y_i}(x_i) > s - 1$. Because of $\Lambda_{M,y_i}(x_i) \in (-\infty, 1]$ for all $i \in [s]$, this is only possible if $\Lambda_{M,y_i}(x_i) > 0$ for all $i \in [s]$. Directly from the definition of $\Lambda_{M,y_i}$ (see also Figure 4), this implies $|x_i - y_i| \leqslant \frac{1}{M}$ for all $i \in [s]$, meaning $x \in y + (M^{-1}[-1,1]^s \times \mathbb{R}^{d-s})$. This proves the first claim.

Regarding the second claim, define $y^* := (y_1, \ldots, y_s) \in \mathbb{R}^s$, and, for $k \in \mathbb{N}$, denote by $\lambda_k$ the Lebesgue measure on $\mathbb{R}^k$. Then, since $[0,1]^d \cap \operatorname{supp} \vartheta_{M,y}^{(s)} \subset (y^* + M^{-1}[-1,1]^s) \times [0,1]^{d-s}$ and $0 \leqslant \vartheta_{M,y}^{(s)} \leqslant 1$, we see that

$$\|\vartheta_{M,y}^{(s)}\|_{L^p([0,1]^d)} \leqslant \left( \lambda_s(y^* + M^{-1}[-1,1]^s) \right)^{1/p} = \left( \frac{2}{M} \right)^{s/p} = 2^{s/p} M^{-s/p}.$$

For the converse estimate, let us also write $x^* = (x_1, \dots, x_s)$ for $x \in \mathbb{R}^d$. Then, if $x \in \mathbb{R}^d$ satisfies $x^* \in y^* + \frac{1}{2Ms}[-1,1]^s$, we see

$$y_i - \frac{1}{M} \leqslant y_i - \frac{1}{2Ms} \leqslant x_i \leqslant y_i + \frac{1}{2Ms} \quad \text{for } i \in [s].$$

By definition of $\Lambda_{M,y_i}$, this implies $\Lambda_{M,y_i}(x_i) = 1 - M \cdot |x_i - y_i| \geqslant 1 - \frac{1}{2s}$ and hence

$$\Delta_{M,y}^{(s)}(x) = \left( \sum_{i=1}^{s} \Lambda_{M,y_i}(x_i) \right) - (s-1) \geqslant s - \frac{1}{2} - (s-1) = \frac{1}{2},$$

so that $\vartheta_{M,y}^{(s)}(x) \geqslant \frac{1}{2}$.

Finally, it is not difficult to show, that

$$\lambda_d\left(\{x \in [0,1]^d : x^* \in y^* + \tfrac{1}{2Ms}[-1,1]^s\}\right) = \lambda_s\left([0,1]^s \cap (y^* + \tfrac{1}{2Ms}[-1,1]^s)\right) \geqslant (2Ms)^{-s},$$

see Grohs & Voigtlaender (2021, Equation (A.1)) for the details. Overall, we thus see

$$\|\vartheta_{M,y}^{(s)}\|_{L^p([0,1]^d)} \geqslant \frac{1}{2} \cdot (2Ms)^{-s/p}. \qquad \square$$

Note that a compactly supported (non-trivial) function such as $\vartheta_{M,y}^{(s)}$ can only be represented by ReLU networks with more than two layers, see Blum & Li (1991, Section 3). For this reason, we focus on the case $L \in \mathbb{N}_{\geqslant 3}$ in this paper. Next, we show that scaled versions of the hat functions $\vartheta_{M,y}^{(s)}$ can be represented using neural networks of a suitable architecture and with a suitable bound on the magnitude of the coefficients. We begin with the (more interesting) case where the exponent $q$ that determines the regularization of the weights satisfies $q \geqslant 2$.

**Lemma A.2.** *Let $d \in \mathbb{N}$, $L \in \mathbb{N}_{\geqslant 3}$, $B \in \mathbb{N}_{\geqslant 3}$, $c > 0$, $q \in [2, \infty]$, and $s \in \mathbb{N}$ with $s \leqslant \min\{\frac{B}{3}, d\}$. Then, there exists a constant*

$$\lambda \geqslant c^L \cdot B^{(L-1)(1-\frac{2}{q})}/12$$

*such that*

$$\nu \cdot \frac{\lambda}{Ms} \cdot \vartheta_{M,y}^{(s)} \in \mathcal{H}_{(d,B,\dots,B,1),c}^q \qquad \forall\, M \in \mathbb{N}, \nu \in \{\pm 1\}, \text{ and } y \in [0,1]^d,$$

*where the $B$ in $(d, B, \dots, B, 1)$ appears $L-1$ times.*

*Proof.* Let $M \in \mathbb{N}$, $y \in [0,1]^d$, and $\nu \in \{\pm 1\}$ be fixed. We will now construct the coefficients $((W^1, b^1), \dots, (W^L, b^L))$ of a neural network with the following properties:

1. The first two layers $((W^1, b^1), (W^2, b^2))$ output at any of their $B$ output dimensions the function $C_1 \cdot \Lambda_{M,y}^{(s)}$ for a suitable scaling factor $C_1 = C_1(c, M, s, B, q) > 0$.

2. The following activation function yields $C_1 \cdot \vartheta_{M,y}^{(s)} = \varrho\left(C_1 \cdot \Lambda_{M,y}^{(s)}\right)$ for all output dimensions.

3. Each of the layers $((W^3, b^3), \dots, (W^{L-1}, b^{L-1}))$ scales the previous output by another factor $C_2 = C_2(c, B, q) > 0$, leading to the output $C_1 C_2^{L-3} \cdot \vartheta_{M,y}^{(s)}$ in any of the $B$ output dimensions. This construction uses the fact that all intermediate outputs are positive by construction such that the intermediate ReLU activation functions $\varrho$ just act as identities.

4. The last layer $(W^L, b^L)$ now computes the sum of the previous outputs scaled by another factor $C_3 = C_3(c, B, q) > 0$ and multiplied by $\nu$, such that the final one-dimensional output equals $\nu B C_1 C_2^{L-3} C_3 \cdot \vartheta_{M,y}^{(s)}$. The result follows by setting $\lambda = B C_1 C_2^{L-3} C_3 Ms$ and choosing the scaling factors $C_1$, $C_2$, and $C_3$ as large as possible, constrained by the width $B$ and the regularization given by $c$ and $q$.

Define $r := \lfloor B/(3s) \rfloor$, noting that $r \geqslant 1$, since $s \leqslant B/3$. We first introduce a few notations: We write $0_{k \times n}$ for the $k \times n$ matrix with all entries being zero; similarly, we write $1_{k \times n}$ for the $k \times n$ matrix with all entries being one. Furthermore, we denote by $(e_1, \ldots, e_d)$ the standard basis of $\mathbb{R}^d$, and define

$$
\begin{aligned}
I_s &:= (e_1 \mid \cdots \mid e_s) \in \mathbb{R}^{d \times s}, \\
\alpha &:= \left( \frac{M^{-1} - y_1}{2} \Big| \frac{M^{-1} - y_2}{2} \Big| \cdots \Big| \frac{M^{-1} - y_s}{2} \right) \in \mathbb{R}^{1 \times s}, \\
\beta &:= \left( -y_1 \mid -y_2 \mid \cdots \mid -y_s \right) \in \mathbb{R}^{1 \times s}, \\
\gamma &:= \left( \tfrac{s-1}{s} \tfrac{1}{2M} \Big| \cdots \Big| \tfrac{s-1}{s} \tfrac{1}{2M} \right) = \frac{s-1}{s} \frac{1}{2M} \cdot 1_{1 \times s} \in \mathbb{R}^{1 \times s}.
\end{aligned}
\tag{5}
$$

We note that all entries of these matrices and vectors are elements of $[-1, 1]$. Using these matrices and vectors, we now define

$$
W^1 := \frac{c}{(3sr)^{1/q}} \Big( \underbrace{I_s/2 | I_s | 0_{d \times s} | \cdots | I_s/2 | I_s | 0_{d \times s}}_{r \text{ blocks of } (I_s/2 | I_s | 0_{d \times s})} \big| 0_{d \times (B - 3rs)} \Big)^T \in \mathbb{R}^{B \times d},
$$

$$
b^1 := \frac{c}{(3sr)^{1/q}} \Big( \underbrace{\alpha \mid \beta \mid \gamma \mid \cdots \mid \alpha \mid \beta \mid \gamma}_{r \text{ blocks of } (\alpha | \beta | \gamma)} \mid 0 \mid \cdots \mid 0 \Big)^T \in \mathbb{R}^B,
$$

and furthermore

$$
W^2 := \frac{c}{(3srB)^{1/q}} \Big( \underbrace{1_{B \times s} \mid -1_{B \times s} \mid -1_{B \times s} \mid \cdots \mid 1_{B \times s} \mid -1_{B \times s} \mid -1_{B \times s}}_{r \text{ blocks of } (1_{B \times s} | -1_{B \times s} | -1_{B \times s})} \mid 0_{B \times (B - 3rs)} \Big) \in \mathbb{R}^{B \times B},
$$

$$
b^2 := (0 \mid \cdots \mid 0)^T \in \mathbb{R}^B,
$$

where we note that $B - 3rs \geqslant 0$ since $r = \lfloor B/(3s) \rfloor$. It is straightforward to verify that $\|W^1\|_{\ell^q}, \|W^2\|_{\ell^q}, \|b^1\|_{\ell^q}, \|b^2\|_{\ell^q} \leqslant c$. Furthermore, we define

$$
W^i := \frac{c}{B^{2/q}} 1_{B \times B} \quad \text{and} \quad b^i := (0 | \cdots | 0)^T \in \mathbb{R}^B \qquad \text{for } 3 \leqslant i \leqslant L - 1,
$$

and finally $W^L := \frac{\nu \cdot c}{B^{1/q}} (1 | \cdots | 1) \in \mathbb{R}^{1 \times B}$ and $b^L := (0) \in \mathbb{R}^1$. Again, it is straightforward to verify that $\|W^i\|_{\ell^q}, \|b^i\|_{\ell^q} \leqslant c$ for $3 \leqslant i \leqslant L - 1$ and also that $\|W^L\|_{\ell^q}, \|b^L\|_{\ell^q} \leqslant c$. Therefore, setting $\Phi := ((W^1, b^1), \ldots, (W^L, b^L))$, we have $R(\Phi) \in \mathcal{H}^q_{(d, B, \ldots, B, 1), c}$; it thus remains to verify that $R(\Phi) = \nu \cdot \frac{\lambda}{Ms} \cdot \vartheta^{(s)}_{M, y}$ for a constant $\lambda$ as in the statement of the lemma.

To see this, we note for any $x \in \mathbb{R}^d$ and $j \in [d]$ that

$$
\begin{aligned}
\varrho\big( \tfrac{x_j}{2} + \tfrac{M^{-1} - y_j}{2} \big) - \varrho(x_j - y_j) &= \tfrac{1}{2} \varrho\big( x_j - y_j + M^{-1} \big) - \varrho(x_j - y_j) \\
&= \begin{cases} 0 & \text{if } x_j \leqslant y_j - M^{-1} \\ \tfrac{1}{2M} \cdot (1 - M \cdot |x_j - y_j|) & \text{if } y_j - M^{-1} < x_j \leqslant y_j \\ \tfrac{1}{2M} \cdot (1 - M \cdot |x_j - y_j|) & \text{if } x_j > y_j \end{cases} \tag{6} \\
&= \tfrac{1}{2M} \Lambda_{M, y_j}(x_j).
\end{aligned}
$$

For notational convenience we further define $\phi^i(x) := \varrho(W^i x + b^i)$ for $i \in [L]$. Then, we observe for $x \in \mathbb{R}^B$ and $i \in [B]$ that

$$
[\phi^2(x)]_i = \frac{c}{(3rsB)^{1/q}} \sum_{b=0}^{r-1} \sum_{j=1}^{s} \Big( x_{3sb+j} - x_{3sb+s+j} - x_{3sb+2s+j} \Big).
$$

Therefore, we see for arbitrary $x \in \mathbb{R}^d$ and $i \in [B]$ that

$$
\begin{aligned}
\big[ (\phi^2 \circ \varrho \circ \phi^1)(x) \big]_i &= \frac{c^2}{(3rs)^{2/q} B^{1/q}} \sum_{b=0}^{r-1} \sum_{j=1}^{s} \Big( \varrho\big( \tfrac{x_j}{2} + \tfrac{M^{-1} - y_j}{2} \big) - \varrho(x_j - y_j) - \varrho\big( \tfrac{s-1}{s} \tfrac{1}{2M} \big) \Big) \\
&= \frac{c^2}{2M(3rs)^{2/q} B^{1/q}} \sum_{b=0}^{r-1} \sum_{j=1}^{s} \Big( \Lambda_{M, y_j}(x_j) - \tfrac{s-1}{s} \Big) \\
&= \frac{c^2 r}{2M(3rs)^{2/q} B^{1/q}} \Delta^{(s)}_{M, y}(x).
\end{aligned}
$$

Hence, it holds that

$$\big(\varrho \circ \phi^2 \circ \varrho \circ \phi^1\big)(x) = \frac{c^2 r}{2M(3rs)^{2/q}B^{1/q}} \cdot \vartheta^{(s)}_{M,y}(x) \cdot (1|\cdots|1)^T \in \mathbb{R}^B.$$

Next, for $3 \leqslant i \leqslant L-1$, we see for arbitrary $\kappa \geqslant 0$ and $j \in [B]$ that

$$\big[\big(\varrho \circ \phi^i\big)\big(\kappa \cdot (1|\cdots|1)^T\big)\big]_j = \varrho\big(\textstyle\sum_{\ell=1}^{B}[W^i]_{j,\ell}\,\kappa\big) = \varrho(cB^{1-\frac{2}{q}}\kappa) = cB^{1-\frac{2}{q}}\kappa,$$

meaning

$$\big(\varrho \circ \phi^i\big)\big(\kappa(1 | \cdots | 1)^T\big) = cB^{1-\frac{2}{q}}\kappa \cdot (1 | \cdots | 1)^T.$$

Therefore, we conclude

$$\big(\varrho \circ \phi^{L-1} \circ \varrho \circ \phi^{L-2} \circ \cdots \circ \varrho \circ \phi^1\big)(x) = \frac{c^{L-1}\,r\,(B^{1-\frac{2}{q}})^{L-3}}{2M(3rs)^{2/q}B^{1/q}}\vartheta^{(s)}_{M,y}(x) \cdot (1 | \cdots | 1)^T \in \mathbb{R}^B.$$

All in all, this easily implies

$$R(\Phi)(x) = \frac{\nu}{B^{1/q}} \sum_{i=1}^{B} \frac{c^L r\,(B^{1-\frac{2}{q}})^{L-3}}{2M(3rs)^{2/q}B^{1/q}}\vartheta^{(s)}_{M,y}(x) = \nu \cdot \frac{c^L\,(B^{1-\frac{2}{q}})^{L-2}(3rs)^{1-\frac{2}{q}}}{6Ms}\vartheta^{(s)}_{M,y}(x).$$

It therefore remains to recall that $r = \lfloor B/(3s)\rfloor \geqslant 1$, so that $2r \geqslant 1 + r > \frac{B}{3s}$ and hence $3rs \geqslant \frac{B}{2}$. Since also $1 - \frac{2}{q} \geqslant 0$, this implies $(3rs)^{1-\frac{2}{q}} \geqslant (B/2)^{1-\frac{2}{q}} \geqslant B^{1-\frac{q}{2}}/2$, which finally shows

$$\lambda := \frac{c^L\,(B^{1-\frac{2}{q}})^{L-2}(3rs)^{1-\frac{2}{q}}}{6} \geqslant \frac{c^L \cdot B^{(L-1)(1-\frac{2}{q})}}{12}. \qquad \square$$

Now, we also consider the case $q \leqslant 2$. We remark that in the case $q = 2$, the next lemma only agrees with Lemma A.2 up to a constant factor. This is a proof artifact and is inconsequential for the questions we are interested in.

**Lemma A.3.** *Let $d \in \mathbb{N}$, $L \in \mathbb{N}_{\geqslant 3}$, $B \in \mathbb{N}_{\geqslant 3}$, $c > 0$, $q \in [1,2]$, and $s \in \mathbb{N}$ with $s \leqslant \min\{d, \frac{B}{3}\}$. Then, we have*

$$\nu \cdot \frac{c^L s^{1-\frac{2}{q}}/(2 \cdot 3^{2/q})}{Ms}\vartheta^{(s)}_{M,y} \in \mathcal{H}^q_{(d,B,\ldots,B,1),c} \quad \forall\, M \in \mathbb{N}, \nu \in \{\pm 1\}, \text{ and } y \in [0,1]^d,$$

*where the $B$ in $(d, B, \ldots, B, 1)$ appears $L - 1$ times.*

*Proof.* The proof idea is similar to the one of Lemma A.2. However, we only realize a scaled version of the function $\vartheta^{(s)}_{M,y}$ in the *first* coordinate of the outputs after the first two layers. As in the proof of Lemma A.2, we denote by $(e_1, \ldots, e_d)$ the standard basis of $\mathbb{R}^d$, and we write $0_{k \times n}$ and $1_{k \times n}$ for the $k \times n$ matrices which have all entries equal to zero or one, respectively. Moreover, we use the matrices and vectors $I_s, \alpha, \beta, \gamma$ defined in Equation (5). With this setup, define

$$W^1 := \frac{c}{(3s)^{1/q}} \cdot \big(I_s/2\,|\,I_s\,|\,0_{d\times(B-2s)}\big)^T \in \mathbb{R}^{B\times d},$$

$$b^1 := \frac{c}{(3s)^{1/q}} \cdot \big(\alpha\,|\,\beta\,|\,\gamma\,|\,0_{1\times(B-3s)}\big)^T \in \mathbb{R}^B.$$

Note that these definitions make sense since $2s \leqslant 3s \leqslant B$. Further, define $b^2 := (0|\cdots|0)^T \in \mathbb{R}^B$ and

$$W^2 := \frac{c}{(3s)^{1/q}} \begin{pmatrix} 1_{1\times s} & -1_{1\times 2s} & 0_{1\times(B-3s)} \\ 0_{(B-1)\times s} & 0_{(B-1)\times 2s} & 0_{(B-1)\times(B-3s)} \end{pmatrix} \in \mathbb{R}^{B\times B}.$$

Next, for $3 \leqslant i \leqslant L-1$, define $b^i := (0|\cdots|0)^T \in \mathbb{R}^B$ and

$$W^i := c \cdot \begin{pmatrix} 1 & 0 & \cdots & 0 \\ 0 & 0 & \cdots & 0 \\ \vdots & \vdots & \ddots & \vdots \\ 0 & 0 & \cdots & 0 \end{pmatrix} \in \mathbb{R}^{B\times B},$$

and finally let $W^L := \nu \cdot c \cdot (1|0| \cdots |0) \in \mathbb{R}^{1 \times B}$ and $b^L := (0) \in \mathbb{R}^1$. It is straightforward to verify that $\|W^j\|_{\ell^q} \leqslant c$ and $\|b^j\|_{\ell^q} \leqslant c$ for all $1 \leqslant j \leqslant L$. Therefore, $R(\Phi) \in \mathcal{H}^q_{(d,B,\dots,B,1),c}$ for $\Phi := ((W^1, b^1), \dots, (W^L, b^L))$. It therefore remains to show that $R(\Phi) = \nu \cdot \frac{c^L s^{1 - \frac{2}{q}} / (2 \cdot 3^{2/q})}{Ms} \vartheta^{(s)}_{M,y}$.

For notational convenience we define $\phi^i(x) := \varrho(W^i x + b^i)$ for $i \in [L]$. Then we note for $3 \leqslant i \leqslant L - 1$ that $\left( \varrho \circ \phi^i \right)(x) = \left( c \cdot \varrho(x_1) \,|\, 0 \,|\, \cdots \,|\, 0 \right)^T$. This easily implies

$$\left( \varrho \circ \phi^{L-1} \circ \varrho \circ \phi^{L-2} \circ \cdots \circ \varrho \circ \phi^3 \right)(x) = \left( c^{L-3} \cdot \varrho(x_1) \,|\, 0 \,|\, \cdots \,|\, 0 \right)^T,$$

and therefore

$$\left( \phi^L \circ \varrho \circ \phi^{L-1} \circ \cdots \circ \varrho \circ \phi^3 \right)(x) = \nu \cdot c^{L-2} \, \varrho(x_1) \quad \text{for } x \in \mathbb{R}^B.$$

Finally, an application of Equation (6) shows that

$$[(\varrho \circ \phi^2 \circ \varrho \circ \phi^1)(x)]_1 = \frac{c^2}{(3s)^{2/q}} \varrho \bigg( \sum_{i=1}^s \left( \varrho\big( \tfrac{x_i}{2} + \tfrac{M^{-1} - y_i}{2} \big) - \varrho(x_i - y_i) - \varrho\big( \tfrac{s-1}{s} \tfrac{1}{2M} \big) \right) \bigg)$$

$$= \frac{c^2}{2M \cdot (3s)^{2/q}} \varrho \bigg( \bigg( \sum_{i=1}^s \Lambda_{M,y_i}(x_i) \bigg) - (s-1) \bigg)$$

$$= \frac{c^2}{2M \cdot (3s)^{2/q}} \varrho(\Delta^{(s)}_{M,y}(x)) = \frac{c^2}{2M \cdot (3s)^{2/q}} \vartheta^{(s)}_{M,y}(x).$$

Overall, we thus see as claimed that

$$R(\Phi)(x) = \nu \cdot c^{L-2} \cdot \frac{c^2}{2M \cdot (3s)^{2/q}} \cdot \vartheta^{(s)}_{M,y}(x) = \nu \cdot \frac{c^L s^{1-\frac{2}{q}} / (2 \cdot 3^{2/q})}{Ms} \cdot \vartheta^{(s)}_{M,y}(x). \qquad \square$$

**Remark A.4.** *A straightforward adaptation of the proof shows that the same statement holds for $\mathcal{H}^q_{(d,B,N_2,\dots,N_{L-1},1),c}$ instead of $\mathcal{H}^q_{(d,B,\dots,B,1),c}$, for arbitrary $N_2, \dots, N_{L-1} \in \mathbb{N}$.*

## A.2 A GENERAL LOWER BOUND

We now show that any target class containing a large number of (shifted) hat functions has a large optimal error.

**Theorem A.5.** *Let $d, m \in \mathbb{N}$, $s \in [d]$, and $M := 8\lceil m^{1/s} \rceil$. Assume that $U \subset C([0,1]^d)$ satisfies*

$$u_0 + \nu \cdot \frac{\lambda}{Ms} \vartheta^{(s)}_{M,y} \in U \qquad \forall \nu \in \{\pm 1\} \text{ and } y \in [0,1]^d$$

*for certain $\lambda > 0$ and $u_0 \in C([0,1]^d)$. Then,*

$$\mathrm{err}^{MC}_m(U, L^p([0,1]^d)) \geqslant \frac{\lambda/4}{(32s)^{1+\frac{s}{p}}} \cdot m^{-\frac{1}{p} - \frac{1}{s}} \qquad \forall p \in [1, \infty].$$

The general idea of the proof is sketched in Section 2.3. In what follows we provide the technical details.

*Proof.* The proof is divided into five steps.

**Step 1:** Define $k := \lceil m^{1/s} \rceil$ and let $y^\ell := \frac{(1,\dots,1)}{8k} + \frac{\ell - (1,\dots,1)}{4k} \in [0,1]^d$ for $\ell \in [4k]^d$. Furthermore, let $\Gamma := [4k]^s \times \{(1,\dots,1)\} \subset [4k]^d$. With

$$f_{\ell,\nu} := u_0 + \nu \cdot \frac{\lambda}{Ms} \vartheta^{(s)}_{M,y^\ell} \quad \text{for} \quad (\ell, \nu) \in \Gamma \times \{\pm 1\}, \tag{7}$$

it holds by assumption that

$$f_{\ell,\nu} \in U \quad \forall (\ell, \nu) \in \Gamma \times \{\pm 1\}. \tag{8}$$

Furthermore, since $M = 8k$, Lemma A.1 and a moment's thought reveal that

$$\forall (\ell, \nu), (\ell', \nu') \in \Gamma \times \{\pm 1\} : \ell \neq \ell' \Rightarrow \mathrm{supp}(f_{\ell,\nu} - u_0)^{\mathrm{o}} \cap \mathrm{supp}(f_{\ell',\nu'} - u_0)^{\mathrm{o}} = \varnothing, \tag{9}$$

where we note that $\text{supp}(f_{\ell,\nu} - u_0) = \text{supp}\,\vartheta_{M,y^\ell}^{(s)}$.

**Step 2:** Let[5] $A \in \text{Alg}_{2m}(U, L^p)$ be arbitrary and $\mathbf{x} = \mathbf{x}(u_0) = (x_1, \ldots, x_{2m}) \in \left([0,1]^d\right)^{2m}$ as described before Equation (1). Put

$$I_{\mathbf{x}} := \left\{ \ell \in \Gamma : \forall\, i \in [2m] : \vartheta_{M,y^\ell}^{(s)}(x_i) = 0 \right\}.$$

We now show that

$$|I_{\mathbf{x}}| \geqslant (4k)^s - 2m. \tag{10}$$

To see this we will estimate the cardinality of the complement set $I_{\mathbf{x}}^c := \Gamma \backslash I_{\mathbf{x}}$ from above. For $\ell \in I_{\mathbf{x}}^c$ there must exist $i_\ell \in [2m]$ with $\vartheta_{M,y^\ell}^{(s)}(x_{i_\ell}) \neq 0$ and hence $x_{i_\ell} \in \text{supp}\left(\vartheta_{M,y^\ell}^{(s)}\right)^\circ$. The map $I_{\mathbf{x}}^c \to [2m]$, $\ell \mapsto i_\ell$, is thus injective due to (9). Therefore $|I_{\mathbf{x}}^c| \leqslant 2m$ and thus $|I_{\mathbf{x}}| \geqslant |\Gamma| - 2m$, which is (10). Furthermore, the definition of $I_{\mathbf{x}}$, combined with the definition of $f_{\ell,\nu}$ in (7) and the condition that $A$ can only depend on the samples $\mathbf{x}$ and the values of the input function at these samples, directly imply that

$$\forall\,(\ell,\nu) \in \Gamma \times \{\pm 1\} : \ell \in I_{\mathbf{x}} \Rightarrow A(f_{\ell,\nu}) = A(u_0). \tag{11}$$

**Step 3:** Recalling our notation for the average in Section 1.2, it holds that

$$\underset{\substack{\ell \in \Gamma \\ \nu \in \{\pm 1\}}}{\sum} \|f_{\ell,\nu} - A(f_{\ell,\nu})\|_{L^p} = \frac{1}{(4k)^s} \sum_{\ell \in \Gamma} \left( \frac{1}{2} \|f_{\ell,-1} - A(f_{\ell,-1})\|_{L^p} + \frac{1}{2} \|f_{\ell,1} - A(f_{\ell,1})\|_{L^p} \right)$$

$$\geqslant \frac{1}{(4k)^s} \sum_{\ell \in I_{\mathbf{x}}} \left( \frac{1}{2} \|f_{\ell,-1} - A(f_{\ell,-1})\|_{L^p} + \frac{1}{2} \|f_{\ell,1} - A(f_{\ell,1})\|_{L^p} \right) \tag{12}$$

$$= \frac{|I_{\mathbf{x}}|}{(4k)^s} \underset{\ell \in I_{\mathbf{x}}}{\sum} \left( \frac{1}{2} \|f_{\ell,-1} - A(f_{\ell,-1})\|_{L^p} + \frac{1}{2} \|f_{\ell,1} - A(f_{\ell,1})\|_{L^p} \right)$$

$$\geqslant \frac{1}{2} \sum_{\ell \in I_{\mathbf{x}}} \left( \frac{1}{2} \|f_{\ell,-1} - A(f_{\ell,-1})\|_{L^p} + \frac{1}{2} \|f_{\ell,1} - A(f_{\ell,1})\|_{L^p} \right) \tag{13}$$

$$= \frac{1}{2} \sum_{\ell \in I_{\mathbf{x}}} \left( \frac{1}{2} \|f_{\ell,-1} - A(u_0)\|_{L^p} + \frac{1}{2} \|f_{\ell,1} - A(u_0)\|_{L^p} \right) \tag{14}$$

$$\geqslant \frac{1}{2} \sum_{\ell \in I_{\mathbf{x}}} \left\| \frac{\lambda}{Ms} \vartheta_{M,y^\ell}^{(s)} \right\|_{L^p} \tag{15}$$

$$\geqslant \frac{1}{2} \cdot (2s)^{-\frac{s}{p}-1} \cdot \lambda \cdot M^{-\frac{s}{p}-1} \tag{16}$$

$$\geqslant \frac{1}{2} \cdot (2s)^{-\frac{s}{p}-1} \cdot \lambda \cdot 16^{-\frac{s}{p}-1} \cdot m^{-\frac{1}{p}-\frac{1}{s}} \tag{17}$$

$$= \frac{\lambda}{2 \cdot (32s)^{\frac{s}{p}+1}} \cdot m^{-\frac{1}{p}-\frac{1}{s}}.$$

Here, (12) follows since $I_{\mathbf{x}} \subset \Gamma$; (13) follows from $k = \lceil m^{\frac{1}{s}} \rceil$ and (10); (14) follows from (11); (15) follows from the triangle inequality and (7); (16) follows from Lemma A.1; and (17) follows from the definition of $M$, which implies that $M \leqslant 8\,m^{1/s} + 8 \leqslant 16\,m^{1/s}$.

**Step 4:** Let $(\mathbf{A}, \mathbf{m}) \in \text{Alg}_m^{MC}(U, L^p)$ be arbitrary with $\mathbf{A} = (A_\omega)_{\omega \in \Omega}$ for a probability space $(\Omega, \mathcal{F}, \mathbb{P})$. Put $\Omega_0 := \{\omega \in \Omega : \mathbf{m}(\omega) \leqslant 2m\}$. Since the Markov inequality implies that

$$m \geqslant \mathbb{E}[\mathbf{m}] \geqslant 2m \cdot \mathbb{P}(\Omega_0^c),$$

it follows that

$$\mathbb{P}(\Omega_0) \geqslant \frac{1}{2}. \tag{18}$$

---

[5]For notational convenience, we abbreviate $L^p([0,1]^d)$ by $L^p$ in this proof.

**Step 5:** We finally estimate for $(\mathbf{A}, \mathbf{m})$ as in Step 4 that

$$\sup_{u \in U} \mathbb{E}\left[\|u - A_\omega(u)\|_{L^p}\right] \geqslant \sum_{\substack{\ell \in \Gamma \\ \nu \in \{\pm 1\}}} \mathbb{E}\left[\|f_{\ell,\nu} - A_\omega(f_{\ell,\nu})\|_{L^p}\right] \tag{19}$$

$$\geqslant \mathbb{E}\left[\mathbb{1}_{\Omega_0}(\omega) \sum_{\substack{\ell \in \Gamma \\ \nu \in \{\pm 1\}}} \|f_{\ell,\nu} - A_\omega(f_{\ell,\nu})\|_{L^p}\right]$$

$$\geqslant \mathbb{P}(\Omega_0) \cdot \frac{\lambda}{2 \cdot (32s)^{\frac{s}{p}+1}} \cdot m^{-\frac{1}{p}-\frac{1}{s}} \tag{20}$$

$$\geqslant \frac{\lambda/4}{(32s)^{\frac{s}{p}+1}} \cdot m^{-\frac{1}{p}-\frac{1}{s}}. \tag{21}$$

Here, (19) follows from (8); (20) follows from Step 3 (note that $A_\omega \in \mathrm{Alg}_{2m}(U, L^p)$ for $\omega \in \Omega_0$); and (21) follows from (18).

Since $(\mathbf{A}, \mathbf{m}) \in \mathrm{Alg}_m^{MC}(U, L^p)$ was arbitrary, this implies the desired statement. $\qquad\square$

**Remark A.6.** *Close inspection of the proof of Theorem 2.2 shows that one can replace the point samples $u(x_i)$ by $Tu(x_i)$, where $T : U \to C([0,1]^d)$ is any local operator[6]. Since any differential operator is a local operator, our lower bounds also hold if we measure point samples of a differential operator applied to $u$, as it is commonly done in the context of so-called physics-informed neural networks (Raissi et al., 2019).*

## B  PROOF OF THE UPPER BOUND IN SECTION 2.4

We first provide an auxiliary result which bounds the spectral norm $\|W\|_{\ell^2 \to \ell^2}$ of a matrix $W$ by its entry-wise $\ell^q$ norm.

**Lemma B.1.** *Let $W \in \mathbb{R}^{N \times M}$ and $q \in [1, \infty]$. Then it holds that*

$$\|W\|_{\ell^2 \to \ell^2} \leqslant \begin{cases} \|W\|_{\ell^q} & \text{if } q \leqslant 2 \\ (\sqrt{NM})^{1-\frac{2}{q}} \cdot \|W\|_{\ell^q} & \text{if } q \geqslant 2. \end{cases}$$

*Proof.* We first note that $\|W\|_{\ell^2} = \|W\|_F$, the Frobenius norm of the matrix $W$. It is well-known that the Frobenius norm satisfies $\|W\|_{\ell^2 \to \ell^2} \leqslant \|W\|_F$. Since we could not locate a convenient reference, we reproduce the elementary proof: The Cauchy-Schwarz inequality implies that

$$\|Wx\|_{\ell^2}^2 = \sum_{i=1}^N \left|\sum_{j=1}^M W_{i,j} x_j\right|^2 \leqslant \sum_{i=1}^N \left(\sum_{j=1}^M |W_{i,j}|^2 \sum_{j=1}^M |x_j|^2\right) = \|W\|_{\ell^2}^2 \cdot \|x\|_{\ell^2}^2,$$

which implies the claim. Thus, we see for $q \leqslant 2$ that $\|W\|_{\ell^2 \to \ell^2} \leqslant \|W\|_{\ell^2} \leqslant \|W\|_{\ell^q}$. Clearly, the same estimate holds for complex-valued matrices and vectors as well.

Now, to handle the case $q \geqslant 2$, we first note for $q = \infty$ and $W \in \mathbb{C}^{N \times M}$ and $x \in \mathbb{C}^M$ that

$$\|Wx\|_{\ell^2}^2 = \sum_{i=1}^N \left|\sum_{j=1}^M W_{i,j} x_j\right|^2 \leqslant \sum_{i=1}^N \left(\sum_{j=1}^M |W_{i,j}|^2 \sum_{j=1}^M |x_j|^2\right) \leqslant \|x\|_{\ell^2}^2 \cdot \|W\|_{\ell^\infty}^2 \cdot NM.$$

This proves the claim in case of $q = \infty$. Finally, for $q \in (2, \infty)$, we choose $\theta = \frac{2}{q}$, so that $\frac{1}{q} = \frac{\theta}{2} + \frac{1-\theta}{\infty}$. Thus, applying the *Riesz-Thorin interpolation theorem* (see, e.g., Folland, 1999, Theorem 6.27) to the linear map $(\mathbb{C}^{N \times M}, \|\cdot\|_{\ell^q}) \to (\mathbb{C}^N, \|\cdot\|_{\ell^2})$, $W \mapsto Wx$, shows for each $x \in \mathbb{C}^M$ that

$$\|Wx\|_{\ell^2} \leqslant (\sqrt{NM})^{1-\theta} \cdot \|W\|_{\ell^q} = (\sqrt{NM})^{1-\frac{2}{q}} \cdot \|W\|_{\ell^q},$$

which completes the proof[7]. $\qquad\square$

---

[6]This means that if $f = g$ on a neighborhood of $x \in [0,1]^d$, then $(Tf)(x) = (Tg)(x)$.

[7]We consider complex matrices and vectors, since the Riesz-Thorin theorem applies as stated only for the complex setting.

Next, let us define the Lipschitz constant $\mathrm{Lip}_{\ell^q}(\phi)$ of a function $\phi : \mathbb{R}^d \to \mathbb{R}^k$ with respect to the $\ell^2$ norm by

$$\mathrm{Lip}_{\ell^q}(\phi) := \sup_{x,y \in \mathbb{R}^d, x \neq y} \frac{\|\phi(x) - \phi(y)\|_{\ell^q}}{\|x - y\|_{\ell^q}}.$$

Note that the Lipschitz constant of an affine-linear mapping $x \mapsto Wx + b$ equals the spectral norm $\|W\|_{\ell^2 \to \ell^2}$. Thus, we can use the previous lemma to bound the Lipschitz constant of neural network realizations $R(\Phi) \in \mathcal{H}^q_{(N_0,\ldots,N_L),c}$ in terms of their architecture $(N_0, \ldots, N_L)$ and the regularization on their weights (given by $\max_{1 \leq i \leq L} \max\{\|W^i\|_{\ell^q}, \|b^i\|_{\ell^q}\} \leq c$).

**Lemma B.2.** *Let $L \in \mathbb{N}$, $q \in [1, \infty]$, $c > 0$, and $N_0, \ldots, N_L \in \mathbb{N}$. Then, each $R(\Phi) \in \mathcal{H}^q_{(N_0,\ldots,N_L),c}$ satisfies*

$$\mathrm{Lip}_{\ell^2}(R(\Phi)) \leq \begin{cases} c^L & \text{if } q \leq 2 \\ c^L \cdot (\sqrt{N_0 N_L} \cdot N_1 \cdots N_{L-1})^{1-2/q} & \text{if } q \geq 2. \end{cases}$$

*Proof.* Let $R(\Phi) \in \mathcal{H}^q_{(N_0,\ldots,N_L),c}$ be arbitrary. By definition, this means

$$R(\Phi) = \phi^L \circ \varrho \circ \phi^{L-1} \circ \cdots \circ \varrho \circ \phi^1,$$

where $\varrho$ acts componentwise, and where the affine-linear maps $\phi^i : \mathbb{R}^{N_{i-1}} \to \mathbb{R}^{N_i}$ are of the form $\phi^i(x) = W^i x + b^i$, with $W^i \in \mathbb{R}^{N_i \times N_{i-1}}$ and $\|W^i\|_{\ell^q} \leq c$.

The ReLU activation function $\varrho : \mathbb{R} \to \mathbb{R}$, $x \mapsto \max\{0, x\}$, is easily seen to satisfy $|\varrho(x) - \varrho(y)| \leq |x - y|$ for $x, y \in \mathbb{R}$. This implies that

$$\mathrm{Lip}_{\ell^2}(R(\Phi)) = \mathrm{Lip}_{\ell^2}(\phi^L \circ \varrho \circ \phi^{L-1} \circ \cdots \circ \varrho \circ \phi^1) \leq \prod_{i=1}^{L} \mathrm{Lip}_{\ell^2}(\phi^i). \tag{22}$$

Lemma B.1 establishes for $i \in [L]$ that

$$\mathrm{Lip}_{\ell^2}(\phi^i) \leq \begin{cases} c & \text{if } q \leq 2 \\ c \cdot (\sqrt{N_{i-1} N_i})^{1 - \frac{2}{q}} & \text{if } q \geq 2, \end{cases}$$

which, together with (22), proves the claim. $\qquad\square$

Note that we can estimate the error of reconstructing Lipschitz continuous functions from samples by piecewise constant interpolation. Together with Lemma B.2, this allows us to construct a (non-adaptive, deterministic) algorithm for reconstructing neural networks from samples.

**Lemma B.3.** *Let $d \in \mathbb{N}$. Then, for every $m \in \mathbb{N}$, there exist points $x_1, \ldots, x_m \in [0,1]^d$ and a map $\Theta_m : \mathbb{R}^m \to L^\infty([0,1]^d)$ satisfying*

$$\left\| \Theta_m\big(u(x_1), \ldots, u(x_m)\big) - u \right\|_{L^\infty([0,1]^d)} \leq \mathrm{Lip}_{\ell^2}(u) \cdot 2\sqrt{d} \cdot m^{-1/d} \tag{23}$$

*for every function $u : [0,1]^d \to \mathbb{R}$ with $\mathrm{Lip}_{\ell^2}(u) < \infty$.*

*Proof.* Let $m \in \mathbb{N}$ be arbitrary and choose $K := \lfloor m^{1/d} \rfloor \geq 1$. Write

$$\{x_1, \ldots, x_{K^d}\} = \tfrac{(1,\ldots,1)}{2K} + \left\{ \tfrac{0}{K}, \tfrac{1}{K}, \ldots, \tfrac{K-1}{K} \right\}^d \quad \text{noting that} \quad [0,1]^d = \bigcup_{i=1}^{K^d} x_i + [-\tfrac{1}{2K}, \tfrac{1}{2K}]^d.$$

Hence, choosing $Q_i := (x_i + [-\tfrac{1}{2K}, \tfrac{1}{2K}]^d) \backslash \bigcup_{j=1}^{i-1}(x_j + [-\tfrac{1}{2K}, \tfrac{1}{2K}]^d)$, we get $[0,1]^d = \biguplus_{i=1}^{K^d} Q_i$, where the union is disjoint.

Note that $K^d \leq m$ and choose arbitrary points $x_{K^d+1}, \ldots, x_m \in [0,1]^d$. Furthermore, define

$$\Theta_m : \quad \mathbb{R}^m \to L^\infty([0,1]^d), \quad (a_1, \ldots, a_m) \mapsto \sum_{i=1}^{K^d} a_i \cdot \mathbb{1}_{Q_i}.$$

To prove Equation (23), let $u : [0,1]^d \to \mathbb{R}$ be arbitrary with $\mathrm{Lip}_{\ell^2}(u) < \infty$. For arbitrary $x \in [0,1]^d$, there then exists a unique $i \in [K^d]$ satisfying $x \in Q_i \subset x_i + [-\frac{1}{2K}, \frac{1}{2K}]^d$, and in particular $\|x - x_i\|_{\ell^2} \leqslant \sqrt{d}/(2K)$. Therefore,

$$\left|\Theta_m\big(u(x_1), \ldots, u(x_m)\big)(x) - u(x)\right| = |u(x_i) - u(x)|$$

$$\leqslant \mathrm{Lip}_{\ell^2}(u) \cdot \|x_i - x\|_{\ell^2} \leqslant \mathrm{Lip}_{\ell^2}(u) \cdot \frac{\sqrt{d}}{2K}.$$

Since $x \in [0,1]^d$ was arbitrary, this implies

$$\left\|\Theta_m(u(x_1), \ldots, u(x_m)) - u\right\|_{L^\infty([0,1]^d)} \leqslant \mathrm{Lip}_{\ell^2}(u) \cdot \frac{\sqrt{d}}{2K}.$$

Finally, we note that $K = \lfloor m^{1/d} \rfloor \geqslant 1$ implies $2K \geqslant 1 + K > m^{1/d}$, which proves the claim. $\square$

Note that the proof above requires to convert a Lipschitz constant with respect to the $\ell_2$ norm to an $\ell_\infty$ estimate which costs a factor $\sqrt{d}$ and contributes to the gap between our lower and upper bound.

**Remark B.4.** *Note that our upper and lower bounds in Theorems 1.1 and 1.4 are asymptotically sharp with respect to the number of samples $m$, the regularization parameter $c$, and the network depth $L$ but not fully sharp with respect to the multiplicative factor depending on $d$ and $q$ only. Given $m$ many samples, a combination of Theorems 1.1 and 1.4 shows that the optimal achievable $L^\infty$ reconstruction error $\varepsilon$ for reconstructing neural networks with $L$ layers up to width $3d$ and coefficients bounded by $c$ in the $\ell^q$ norm satisfies*

$$\begin{cases} \frac{1}{256 \cdot 3^{2/q}} \cdot c^L \cdot d^{-\frac{2}{q}} \cdot m^{-\frac{1}{d}} \\ \frac{1}{1536 \cdot d} \cdot c^L \cdot (3d)^{(L-1)(1-\frac{2}{q})} \cdot m^{-\frac{1}{d}} \end{cases} \leqslant \varepsilon \leqslant \begin{cases} \sqrt{d} \cdot c^L \cdot m^{-\frac{1}{d}} & \text{if } q \leqslant 2 \\ d^{1-\frac{1}{q}} \cdot c^L \cdot (3d)^{(L-1)(1-\frac{2}{q})} \cdot m^{-\frac{1}{d}} & \text{if } q > 2. \end{cases}$$

*For moderate input dimensions $d$ the upper and lower bounds are quite tight, but for larger $d$ there remains a gap. However, in that case the lower bound for $m$ is already intractable (at least if $\varepsilon \ll 1/d$ or if $c \gg 1$ and $L$ is large) so that the upper bound is merely of academic interest.*

# C  Hyperparameters used in the numerical experiments

Table 1: General hyperparameters for the experiments in Figure 1 and Section 3.

| Description | Value | Variable |
|---|---|---|
| **Experiment** | | |
| precision | float64 | |
| GPUs per training | 1 (NVIDIA GTX-1080, RTX-2080Ti, A40, or A100) | |
| **Deep learning algorithms** | | |
| optimizer | Adam | |
| initialization of coefficients $(W^\ell, b^\ell)$ | $\mathcal{U}([-\sqrt{1/N_{\ell-1}}, \sqrt{1/N_{\ell-1}}])$ | |
| activation function | ReLU | $\varrho$ |
| learning rate scheduler | exponential decay | |
| initial / final learning rate | $10^{-4}$ / $10^{-6}$ | |
| decay frequency | every epoch | |
| **Evaluation** | | |
| number of samples | $2^{24}$ | $J$ |
| distribution of samples | $\mathcal{U}([-0.5, 0.5]^d)$ | |
| evaluation norm | $\{1, 2, \infty\}$ | $p$ |
| number of evaluations | 5 (evenly spaced over all epochs) | |

Table 2: Hyperparameters specific to the experiment in Figure 1.

| Description | Value | Variable |
|---|---|---|
| **Experiment** | | |
| samples | $10^3$ | $m$ |
| dimension | 1 | $d$ |
| **Target function** | | |
| sinusoidal function | $x \mapsto \log(\sin(50x) + 2) + \sin(5x)$ | $u$ |
| **Deep learning algorithm** | | |
| depth of architecture | 22 | $L$ |
| width of architecture | 50 | $N_1, \ldots, N_{L-1}$ |
| batch-size | 20 | |
| number of epochs | 5000 | |

Table 3: Hyperparameters specific to the experiments in Section 3.

| Description | Value | Variable |
|---|---|---|
| **Experiment** | | |
| samples | $\{10^2, 10^3, 10^4, 10^5\}$ | $m$ |
| dimension | $\{1, 3\}$ | $d$ |
| **Target functions** | | |
| number of teachers | 40 | $|\widehat{U}|$ |
| depth of teacher architecture | 5 | $L$ |
| width of teacher architecture | 32 | $N_1, \ldots, N_{L-1}$ |
| activation of teacher | ReLU | $\varrho$ |
| teacher coefficient norm | $\infty$ | $q$ |
| teacher coefficient norm bound | 0.5 | $c$ |
| distribution of coefficients | $\mathcal{U}([-0.5, 0.5])$ | |
| **Deep learning algorithms** | | |
| number of seeds | 3 | $|\widehat{\Omega}|$ |
| depth of student architecture | 5 | $L$ |
| width of student architecture | $\{32, 512, 2048\}$ | $N_1, \ldots, N_{L-1}$ |
| batch-size | $\{100, m/5, m/50\}$ | |
| number of epochs | 500 (if batch-size = 100), 5000 (else) | |

