# OpenReview forum: "Learning ReLU networks to high uniform accuracy is intractable"
_ICLR.cc/2023/Conference — ICLR 2023 poster_

### Official Review · Reviewer_zxQe · 2022-10-21

**Confidence:** 4
**Clarity, Quality, Novelty And Reproducibility:** (see Strengths and Weaknesses)
**Correctness:** 4
**Technical Novelty And Significance:** 4
**Empirical Novelty And Significance:** 3
**Recommendation:** 8

**Strength And Weaknesses:**

### Strengths

- The paper considers an unorthodox (but important and impactful) statistical learning problem: sample complexity for uniform accuracy (instead of expected accuracy). The motivation and impact of such an analysis are discussed carefully throughout the manuscript. The main ideas and techniques of the work were developed based on well-founded rationales and in general, the paper is well-written with a sufficient literature review.
- One of the important (and maybe novel) insights of the work about neural networks in this learning setting is that while there exist functions that can be exactly represented by small neural networks, these representations cannot be inferred from samples because of the network’s expressiveness.
- The mathematical analyses of the work are rigorous, intuitive, and seem correct.
- One main technical contribution of the work is to explicitly show that neural network models contain a large class of “bump functions” with estimable L^p norms and disjoint supports. Thus, with a “small” number of observations, no algorithm would be able to distinguish those functions from the zero function, or one another. The sample complexity depends on p, and when p=\infty (uniform accuracy), the dependency is exponential.
- Beyond this direct application of the analysis, several components of its explicit computations would be of great interest to a broader audience. For example, the bump functions, which have support in B(y, 1/M) and are greater than 1/2 in B(y, 1/(2Ms)), are very similar in spirit to the concept of mollifiers and approximation of the Delta Dirac function. The fact that they can be represented by a standard feed-forward ReLU network could be a useful tool in other contexts.

### Weaknesses

- None noted.

### Additional comments

- As stated in the comments, I think the explicitness of Lemmas A.1 and A.2 are technically interesting, and some detailed descriptions of the insights of those constructions would be appreciated. For example, the steps (and the constructions of the layers) in the proof of Lemma A.2 have their own meaning (representing the hat functions, representing the bump functions, controlling the norms), and some verbal descriptions of what they do would make the parts more accessible. (I want to note that it’s just an optional comment and will not affect my review decision.)

**Summary Of The Paper:**

The paper aims to show that for any learning problem formulated over target classes containing ReLU neural networks of a prescribed architecture, the number of training samples needed for an adaptive randomized algorithm to guarantee a given uniform accuracy scales exponentially both in the depth and the input dimension of the network architecture.

**Summary Of The Review:**

The paper rigorously addresses a meaningful question in applied machine learning. The work provides novel insights into the learning problem, and its technical contributions might be of broad interest.

---

> ### Author Response · Authors · 2022-11-19
> **Response to reviewer zxQe**
>
> We thank you for your very positive feedback and are very happy that you appreciate our work.
>
> > **As stated in the comments, I think the explicitness of Lemmas A.1 and A.2 are technically interesting, and some detailed descriptions of the insights of those constructions would be appreciated. For example, the steps (and the constructions of the layers) in the proof of Lemma A.2 have their own meaning (representing the hat functions, representing the bump functions, controlling the norms), and some verbal descriptions of what they do would make the parts more accessible. (I want to note that it’s just an optional comment and will not affect my review decision.)**
>
> Thank you for the suggestions. We agree with your assessment and added corresponding verbal explanations of our proof strategies to Lemmas A.1, A.2, and A.3 in the appendix.

---

### Official Review · Reviewer_eNk7 · 2022-10-24

**Confidence:** 3
**Correctness:** 4
**Technical Novelty And Significance:** 3
**Empirical Novelty And Significance:** 3
**Recommendation:** 6

**Clarity, Quality, Novelty And Reproducibility:**

The paper builds off various works on expressivity and learning of deep
feedfoward neural networks to establish a new result, with a motivation that
seems to also be new. The work is clearly written and well-referenced.
Mathematical claims are very clearly written, without ambiguous notation.



**Strength And Weaknesses:**

## Strengths

- The lower bound holds very generally and without conditions associated with
  other SQ-type lower bounds that apply to more natural algorithms like GD/SGD.
- The focus on $L^\infty$ estimation has possible connections to
  security-critical settings where one is interested in the ability to recover
  the parameters of a neural network model.
- The paper is very clearly written and well-referenced. It is possible to
  understand the paper's main results at a technical level without having to
  reference notation/etc. in the appendices because background information is
  explained completely.

## Weaknesses

Although I have not checked the proofs, the authors' results seem reasonable to
me. Therefore most of my comments here will not be weaknesses with the
(well-written) paper per se, but rather questions about the motivation and
implications for practice of the theory.

- The requirement $B \leq 3d$ in Theorem 2.2 seems unnatural -- can this be
  improved? Is there a "functional" description (e.g. from the proof) of what
  this constraint implies?
- The exponential dependence on dimension may be seen as not too
  surprising: one needs a number of points exponential in dimension just to
  cover the input space, and the class seems complex enough that it is really
  necessary to have such a dense covering. What seems more interesting is
  that it is also necessary to sample exponential in depth. However, one is
  naturally curious about the extent to which this is simply because the
  learning problem is extremely hard -- arbitrary depth-L networks can
  parameterize very complex functions, with many linear regions (e.g.
  following Montúfar's work) -- and whether the same exponential dependencies
  would appear for learning e.g. a Sobolev or a Besov class. Another angle
  on this question would be to consider how the bound would look for learning a
  "smoothed" class of ReLU networks, e.g. those with a bounded number of
  linear regions, or the like. The behavior that may be most interesting as a
  theory for the practical successes of deep learning would be one that
  captures the ability of deeper networks to *adaptively* represent complex
  functions, when it is necessary, given sufficient samples -- and it seems
  this kind of explanation is out of reach of the present theory, given that
  the class being learned contains extremely complex functions implemented by
  general depth-$L$ networks. (The authors mention this as a direction for
  future work in the conclusion; it would be nice to hear some thoughts in this
  context in the response.)
- Similarly to the previous point, I am not completely convinced about the
  connection the authors make between their result and adversarial robustness
  -- although indeed in the setting of adversarial robustness one is interested
  in $L^\infty$ stability guarantees, these are generally not in the
  teacher-student setting where the training labels are generated by some deep
  neural network, but rather by some dataset with certain structures
  (statistical, geometric, etc.). In this latter setting, the requirements to
  be robust to worst-case corruptions may be quite different from the setting
  that is considered here. I think this is especially pertinent with regards to
  the $\exp(L)$ dependence in the authors' lower bound -- how would the
  corresponding lower bound look in the setting of a structured, labelled
  dataset?


## Minor / Questions / Comments
- Is there a version of Theorem 1.1 that applies to networks of arbitrary
  intermediate widths (like Theorem 1.4)?
- In Theorem 1.1, when $q \leq 2$ the result seems to be vacuous in high
  dimensions unless $\varepsilon$ goes to zero; on the other hand, it seems
  that working through the corresponding bound in Theorem 2.2 implies a
  nonvacuous result in high dimensions (it seems the dependence on width $B$
  has been worst-cased to a dimension dependence in Theorem 1.1). Would it
  make sense to state the result differently? Reading Theorem 1.1, I have the
  sense that the bound is very loose with respect to Theorem 1.4 for "mild
  regularization" (whereas with the actual dependence, the gap between the
  bounds becomes much clearer).
- Some comments on *what the class of functions implemented by the
  architectures considered is* would be helpful here in assessing the degree
  of difficulty of the learning problem, relative to some of the other
  classes mentioned in Remark 1.3 -- e.g. in the proof idea section describing
  the bump function, can we also know Sobolev estimates for these functions (to
  better understand what the class being embedded is, effectively)?
  - The proof sketch of the lower bound reminds of some of the description of
  Bubeck and Sellke's work on robustness is there any connection here?
  (Keywords like Baum's construction of memorizing networks and its
  generalization by Bubeck et al., etc...)
- Could the authors provide a proof sketch for the upper bound in Theorem 2.4?
  Given that this upper bound applies to an adaptive, potentially randomized
  algorithm, is it safe to assume that this upper bound applies to an algorithm
  that is "stronger" than e.g. empirical risk minimization under a
  distributional assumption, which as the authors mention in related work has
  been considered in works with a similar flavor? I am just interested in how
  to compare this upper bound to results I have seen in related work on $L^p$
  $p \ll +\infty$ learning of ReLUs with certain algorithms.
- It might be more correct to say that the "minimal error over neural network
  target functions" is estimated/approximated, rather than "computed", on page 8.


**Summary Of The Paper:**

The authors study the problem of learning classes of deep, feedfoward neural
networks with ReLU activations through with guarantees in terms of $L^\infty$
accuracy, through the angle of sample complexity requirements. More precisely,
they consider a rather general model where a learner can adaptively query
points of the solid cube $[0, 1]^d$ for evaluations of an unknown deep ReLU
network $f$, and provide lower bounds on the number of queries necessary for
any (possibly randomized) such algorithm to output an $\varepsilon$
approximation to $f$, for all possible choices of the network $f$ in a
restricted class (bounded number of neurons, bounded depth, and norm-bounded
weights). The obtained lower bound grows exponentially in the dimension $d$ and
the network depth, demonstrating a strong barrier to learning in this setting.
Upper bounds are derived in the same algorithmic setting (as before,
computational complexity is not considered)



**Summary Of The Review:**


The paper presents a solid contribution to the understanding of teacher-student
learning of neural networks to a high degree of uniform accuracy. I am unsure
of the general implications of the theory given the student-teacher setting,
but I think the work may lead to many interesting follow-up investigations, as
the authors mention in the conclusion.

---

> ### Author Response · Authors · 2022-11-19
> **Response to reviewer eNk7 (1/4)**
>
> We appreciate your detailed review and the positive feedback. Please find our answers to your suggestions and questions below.
>
> >
> > **The requirement $B\le 3d$ in Theorem 2.2 seems unnatural -- can this be improved? Is there a "functional" description (e.g. from the proof) of what this constraint implies?**
>
> The point of the statement in Theorem 1.1 is that even though the considered network class is “not too complicated” (since we only consider networks with up to $3d$ neurons per layer), we still get the stated lower bound for the required number of samples. We also want to emphasize that our theorem considers the set of *all* neural networks with width up to $3d$ in each of the hidden layers.
>
> In Theorem 2.2, which you are referring to, the width $B\in \mathbb{N}$ can actually be chosen arbitrary. The condition $s\le \min (\frac{B}{3}, d)$ arises from the fact that with $3 s$ neurons in each layer (in particular in the first layer), we are able to implement an essentially "$s$-dimensional bump function" (called $\vartheta\_{M,y}^{(s)}$, see Section A.1), where naturally $s$ cannot be larger than the input dimension $d$. It should be noted, however, that if $B$ is very large (i.e., larger than $3d$), this does improve the lower bound in the case where $q > 2$, i.e., whenever the boundedness condition imposed on the network weights is not too severe. If, however, $q \leq 2$, then increasing the width beyond $s = 3d$ does not increase the lower bound. This is inevitable in view of the upper bound (Theorem 2.4).

---

> ### Author Response · Authors · 2022-11-19
> **Response to reviewer eNk7 (2/4)**
>
> >
> > **The exponential dependence on dimension may be seen as not too surprising: one needs a number of points exponential in dimension just to cover the input space, and the class seems complex enough that it is really necessary to have such a dense covering. What seems more interesting is that it is also necessary to sample exponential in depth. However, one is naturally curious about the extent to which this is simply because the learning problem is extremely hard -- arbitrary depth-L networks can parameterize very complex functions, with many linear regions (e.g. following Montúfar's work) -- and whether the same exponential dependencies would appear for learning e.g. a Sobolev or a Besov class. Another angle on this question would be to consider how the bound would look for learning a "smoothed" class of ReLU networks, e.g. those with a bounded number of linear regions, or the like. The behavior that may be most interesting as a theory for the practical successes of deep learning would be one that captures the ability of deeper networks to adaptively represent complex functions, when it is necessary, given sufficient samples -- and it seems this kind of explanation is out of reach of the present theory, given that the class being learned contains extremely complex functions implemented by general depth-$L$ networks. (The authors mention this as a direction for future work in the conclusion; it would be nice to hear some thoughts in this context in the response.)**
>
> Thank you for this comment. We do not entirely agree with the intuition that "the class seems complex enough that it is really necessary to have such a dense covering": in fact, the class can be (continuously) parametrized by a number of parameters that does *not* scale exponentially in the dimension (in more technical terms and in view of the boundedness of the weights: the metric entropy does *not* scale exponentially in the dimension)! We are not aware of any other interesting class that satisfies such a gap between metric entropy and sample complexity. In particular, the mentioned Sobolev and Besov classes do *not* exhibit this gap (i.e., for these classes, the required number of samples scales exponentially in the dimension, but so do the entropy numbers; the notion of depth does not apply for these classes)! The intuitive reason for the exponential dependence on the depth is that the magnitude of activations can scale exponentially in the depth and thus allows us to represent more localized bump functions with the same parameter constraint. We agree that studying the algorithmic complexity of deep learning deserves more research and the directions outlined by the reviewer are certainly interesting. Regarding the idea of considering ReLU networks with a bounded number of linear regions, we note that an unbounded number of linear regions is already necessary for being able to represent smooth functions (see for example [https://arxiv.org/abs/1709.05289](https://arxiv.org/abs/1709.05289)); furthermore, it seems very difficult to impose a bound on the number of linear regions in the training of neural networks (or in fact in any other concrete algorithm). What might be interesting is to show that the neural networks obtained via (S)GD exhibit a more benign sampling behaviour. One possibility for doing so might be to utilize the recent work on the implicit bias imposed by (S)GD. We feel, however, that getting a meaningful result in this direction will be extremely difficult.
>
>
>
> >
> > **Similarly to the previous point, I am not completely convinced about the connection the authors make between their result and adversarial robustness -- although indeed in the setting of adversarial robustness one is interested in stability guarantees, these are generally not in the teacher-student setting where the training labels are generated by some deep neural network, but rather by some dataset with certain structures (statistical, geometric, etc.). In this latter setting, the requirements to be robust to worst-case corruptions may be quite different from the setting that is considered here. I think this is especially pertinent with regards to the $\exp(L)$ dependence in the authors' lower bound -- how would the corresponding lower bound look in the setting of a structured, labelled dataset?**
>
> We agree to some extent with the assessment regarding the direct application of our results to adversarial robustness and reformulated our text in the discussion section accordingly. Our results show, roughly speaking, that knowledge of labels on a number of samples that does not scale exponentially on $L$ and $d$ does not carry sufficient information to determine a neural network up to small *uniform* error.

---

> ### Author Response · Authors · 2022-11-19
> **Response to reviewer eNk7 (3/4)**
>
> >
> > **Is there a version of Theorem 1.1 that applies to networks of arbitrary intermediate widths (like Theorem 1.4)?**
>
> Yes, as described above, our general Theorem 2.2 (referenced after Theorem 1.1) applied to arbitrary widths $B\in\mathbb{N}$. One could also easily extend our proof strategy to different widths per layer (see also Remark A.4). However, this gets quite technical and we believe that there is not much additional insight.
>
> >
> > **In Theorem 1.1, when $q\le 2$ the result seems to be vacuous in high dimensions unless $\varepsilon$ goes to zero; on the other hand, it seems that working through the corresponding bound in Theorem 2.2 implies a nonvacuous result in high dimensions (it seems the dependence on width $B$ has been worst-cased to a dimension dependence in Theorem 1.1). Would it make sense to state the result differently? Reading Theorem 1.1, I have the sense that the bound is very loose with respect to Theorem 1.4 for "mild regularization" (whereas with the actual dependence, the gap between the bounds becomes much clearer).**
>
> You are correct in noting that the statement of Theorem 1.1 for $q \leq 2$ only implies an interesting bound if $\epsilon$ is small compared to $1/d$. As you noted, this occurs because we tried to phrase the result as simple as possible. The more technical statement in Theorem 2.2 still yields non-trivial results for $q \leq 2$ also for moderately large $\epsilon$. We deliberately chose this formulation to make Theorem 1.1 as easy to understand as possible and prefer to keep it this way. If you feel strongly about this point, we are willing to reformulate Theorem 1.1 so as to make it closer to Theorem 2.2 for the camera-ready version.
>
> >
> > **Some comments on what the class of functions implemented by the architectures considered is would be helpful here in assessing the degree of difficulty of the learning problem, relative to some of the other classes mentioned in Remark 1.3 -- e.g. in the proof idea section describing the bump function, can we also know Sobolev estimates for these functions (to better understand what the class being embedded is, effectively)?**
>
> All the mentioned classes, i.e., neural networks (with *arbitrary architectures*), polynomials (of arbitrary degree), and RKHS are universal (under mild conditions) in the sense that they can approximate any continuous function on any compact set arbitrarily well. However, it is still an active area of research to characterize the class of functions represented by neural networks with *fixed architecture* (and regularized coefficients) and we refer to [Gribonval et al. (2019)](https://arxiv.org/abs/1905.01208). For classical smoothness spaces, e.g., Sobolev or Besov spaces, there exist a series of approximation results, quantifying the approximation error in terms of the size of the architecture, see, for instance, [Gühring et al. (2019)](https://arxiv.org/abs/1902.07896), showing that neural networks achieve the same optimal rates as polynomials. However, it is known that neural networks can also efficiently approximate highly oscillating functions or other approximation spaces such as Wavelets or Gabor frames, see [Elbrächter et al. (2019)](https://arxiv.org/abs/1901.02220). Thus, neural networks are more "flexible" in the sense that they can efficiently approximate a variety of function spaces. In the other direction, from Lemma A.2 it can be seen that the Lipschitz constant (and therefore also the $W^{1,\infty}$-Sobolev norm) of the considered neural networks (and thus of the considered bump functions) can be bounded, but the exponential (in $d$) dependence of the sample complexity still holds true.
>
> >
> > **The proof sketch of the lower bound reminds of some of the description of Bubeck and Sellke's work on robustness is there any connection here? (Keywords like Baum's construction of memorizing networks and its generalization by Bubeck et al., etc...)**
>
> We suspect that you are referring to the work https://proceedings.neurips.cc/paper/2020/hash/34609bdc08a07ace4e1526bbb1777673-Abstract.html. On a superficial level there is indeed a similarity in the sense that both works study the expressivity of samples evaluated on classes of neural networks. However, on closer inspection the two results and techniques are quite different, and, in fact, in a sense dual to each other: While in Bubeck et al.'s work the aim is to find a single neural network whose samples are equal to a given set of labels, our aim is to study the width of all neural networks whose samples yield equal values.

---

> ### Author Response · Authors · 2022-11-19
> **Response to reviewer eNk7 (4/4)**
>
>
> >
> > **Could the authors provide a proof sketch for the upper bound in Theorem 2.4? Given that this upper bound applies to an adaptive, potentially randomized algorithm, is it safe to assume that this upper bound applies to an algorithm that is "stronger" than e.g. empirical risk minimization under a distributional assumption, which as the authors mention in related work has been considered in works with a similar flavor? I am just interested in how to compare this upper bound to results I have seen in related work on $L^p\ p \ll +\infty$ learning of ReLUs with certain algorithms**
>
> We already had some explanation of the results in the appendix. However, we agree that also the main text should convey the main idea of the proof and we added this to the revised version, immediately after the statement of Theorem 2.4. The reconstruction algorithm is actually very simple (i.e., the possible randomness and adaptivity does not need to be used).
>
> >
> > **It might be more correct to say that the "minimal error over neural network target functions" is estimated/approximated, rather than "computed", on page 8.**
>
> Thank you for this suggestion, we changed it accordingly in the revised version.

---

> > ### Comment · Reviewer_eNk7 · 2022-12-06
> > **thanks**
> >
> > I thank the authors for their patient and illuminating response, and for resolving some minor issues and additions to the submission. Here are a few specific points I would like to mention:
> > - [Regarding arbitrary widths $B$] Unless I am still missing something, I think it would be useful to have the simplified version of the main theorem stated in a way that suggests there are no constraints on the widths required for it to hold. I have the feeling that readers that are slightly out-of-area (not working on approximation theory) may appreciate this as a suggestion that the authors' results are fully general in this sense. I agree about varying intermediate widths.
> > - [The $q \leq 2$ simplifications in Theorem 1.1] No, I think the way you have stated it is more than sufficient. This concern only occurred to me while reading due to some comments about tightness of the lower bound made later in the submission (I think it was the second graf of the limitations section), and perhaps adding a specific `\ref` to the precise theorems (not the simplified ones, which is where I went to look after reading that graf) would eliminate this confusion.
> > - [The exponential dependence on depth] Thank you for this remark -- I agree this is a very interesting technical issue.

---

> > > ### Author Response · Authors · 2022-12-09
> > > **Second response to reviewer eNk7**
> > >
> > > We are very pleased that our response and revision was helpful and successfully addressed certain issues. We also thank you for your further suggestions.
> > >
> > > In the camera-ready version, we will put more emphasis on our general results. In particular, we will state in Theorem 1.1 that the lower bound applies to arbitrary widths, and we will again refer to the general bounds in the "discussion and limitations" section.
> > >
> > > Finally, we are happy to answer any remaining questions. Given our "illuminating" response and our revision that resolved some minor issues, we also hope that you may consider updating your recommendation.

---

### Official Review · Reviewer_Jti4 · 2022-10-28

**Confidence:** 4
**Correctness:** 4
**Technical Novelty And Significance:** 2
**Empirical Novelty And Significance:** 1
**Recommendation:** 5

**Clarity, Quality, Novelty And Reproducibility:**

I have some questions that are not clear from the paper:
- Why does the width need a dependence on $d$? What is the actual dependence on the width independent of $d$?
- The construction of these bump functions uses only 2-layers to actually construct the “hat” function and then the rest of the layers are pass through layers that affect only the scaling of the function (hence the Lipschitz constant). Similarly, the width shows up in its interplay with the norm constraint affecting the scaling. Assuming that the $\ell_1$-norm of the weights is bounded by 1 at each layer would remove this blow-up. Is there any use of depth outside of this?
- What are the differences with the constructions in Telgarsky 2015? I understand that the work deals with purely approximation and not sample complexity, but they do bound the total error in the interval.

**Strength And Weaknesses:**

Strengths:
- The paper makes progress towards understanding the information theoretic limits of robust learning, which is an important problem in the light of adversarial vulnerabilities in current ML models. The lower bounds hold for all algorithms and not restricted to some computational models.
- The paper is well-written and easy to follow. The authors do a good job of thoroughly discussing the theorem statements and their implications.

Weaknesses:
- I am not convinced by the technical depth of the main result and more generally the setting. It is worst-case in nature and for a very general class. This makes the lower bound weaker, since getting $\ell_\infty$ bounded error in such a setting seems naturally hard. The examples in Remark 1.3 where such a goal is tractable are restricted to polynomials/linear functions which have special structure in terms of interpolation.
- Experiments are in very low dimensions. Trends for different errors also seem similar. Hence, it is unclear what the reader should take away from the experiments.

**Summary Of The Paper:**

Motivated by robustness of trained models, the paper studies the statistical complexity of learning the class of ReLU networks up to $\ell_\infty$-error (uniform error). In particular, they show that the sample complexity required for learning ReLU neural networks with $L $ layers each with width at most $3d$ (where $d$ is the input dimension) and bounded norm coefficients requires exponential in $d$ and $L $ samples. In contrast, learning to $\ell_2$-error requires polynomially on $d$. They also give (almost) matching upper bounds. The results in the paper allow for the underlying algorithm to be adaptive in the selection of the data points, that is, it can query the points necessary to give the required guarantee. The key idea is to use the flatness of ReLU to show that the NN class contains functions of the following form: for any $y\in \mathbb{R}^d$ the function is non-zero only for a small ball around point $y$ in the space. Therefore, to learn the function from samples to $\ell_\infty$ accuracy, one needs to sample a point in every small ball, that is, cover the entire space which requires exponential in $d $ samples. The norm bounds on the weights governs the radius of these balls and the base of the exponent.

**Summary Of The Review:**

Overall, in my opinion, the worst-case nature of the result makes it not very interesting. The requirement of $\ell_\infty$-bounded error becomes extremely strong, and not very insightful for realistic models. As the authors point out in the limitations, studying this in a more reasonable sub-class of neural networks or understanding uniform error restricted to some manifold or under some perturbation model would be more useful and interesting.

Post rebuttal: Increased the score from 3-> 5.

---

> ### Author Response · Authors · 2022-11-19
> **Response to reviewer Jti4 (1/2)**
>
> We thank you for your review and the feedback. Let us comment on your questions below.
>
> >
> > **I am not convinced by the technical depth of the main result and more generally the setting. It is worst-case in nature and for a very general class. This makes the lower bound weaker, since getting bounded error in such a setting seems naturally hard. The examples in Remark 1.3 where such a goal is tractable are restricted to polynomials/linear functions which have special structure in terms of interpolation.**
>
> While it is probably true that linear functions are very special, so that it is not surprising that they can be learned to uniform accuracy with a reasonable number of samples, this is much less clear for sparse polynomials (for which decidedly non-trivial results from compressive sensing are needed) and for certain classes of RKHS (see Remark 1.3) which rely on recent progress based on the solution of the Kadison Singer problem. Thus, there are interesting classes of functions for which learning them to uniform accuracy is tractable. We therefore don't agree with the assessment that "getting bounded error in such a setting seems naturally hard". Moreover, we would appreciate more concrete feedback as to why the reviewer is "not convinced by the technical depth of the main result". The property that uniform accuracy is intractable is in fact highly special to neural network classes and does not hold for most other model classes of interest. We therefore believe that pointing this out and formally proving it is an important contribution to the literature.
>
> >
> > **Experiments are in very low dimensions. Trends for different errors also seem similar. Hence, it is unclear what the reader should take away from the experiments.**
>
> Our code can readily be executed for any dimension. However, we already executed 8640 experiments for these plots and are unfortunately restricted by our computational resources. Moreover, we face the severe issue that *the uniform accuracy cannot be reliably evaluated in higher dimensions*. To sample every $\epsilon$-neighborhood in $d$ dimensions, one would need $\epsilon^{-d}$ many samples which is already infeasible for small dimension. For our experiments, we already use $2^{24}$ samples to obtain a good estimate for dimensions $3$ and $5$. The plots provide empirical evidence for the size of the gap between $p=1,2$ and $p=\infty$. Regarding the trends (scaling between error and sample size), the experiments are indeed similar. There are two reasons for this. First, the differences in the trend (as predicted by our theory) becomes more pronounced as the dimension increases and for the low dimensions considered in the experiments the theoretically predicted trends are also quite close to each other. Second, our algorithmic setting can only approximately evaluate the optimal rate because it is impossible to test all conceivable algorithms. What is clearly observed is that the error for $L^\infty$ approximation is significantly larger than for $L^1$ or $L^2$ approximation.
>
> >
> > **Why does the width need a dependence on $d$? What is the actual dependence on the width independent of $d$?**
>
> The dependence on the width only appears in Theorem 1.1. The more general result in Theorem 2.2 (referenced after Theorem 1.1) applies to arbitrary widths $B\in\mathbb{N}$. One could also easily extend our proof strategy to different widths per layer (see also Remark A.4). However, this gets quite technical and we believe that there is not much additional insight.
>
> If the width would be fixed independent of $d$ the resulting hypothesis class would (for large $d$) not be capable of representing any truly high dimensional function, since a narrow layer would project the input onto a low dimensional subspace.
>
> >
> > **The construction of these bump functions uses only 2-layers to actually construct the “hat” function and then the rest of the layers are pass through layers that affect only the scaling of the function (hence the Lipschitz constant). Similarly, the width shows up in its interplay with the norm constraint affecting the scaling. Assuming that the $\ell^1$-norm of the weights is bounded by 1 at each layer would remove this blow-up. Is there any use of depth outside of this?**
>
> The reviewer is correct and the statement regarding the removal of the blow-up in $L$ is contained in our results (for $q=2$, which corresponds to a bounded Hilbert-Schmidt norm and hence Lipschitz constant of each layer). Note that our upper bound shows that, to some extent, our construction is optimal, i.e., using the deeper layers in a "more clever way" would not yield a significantly stronger lower bound.

---

> > ### Comment · Reviewer_Jti4 · 2022-11-21
> > **Response to Rebuttal**
> >
> > Thanks to the authors for a detailed response to my comments, and addressing several concerns. I am willing to increase my score, but I still have some concerns. Here are some lingering comments/concerns:
> >
> > > We would be curious to know which of our claims you think have minor issues such that we could fix those.
> >
> > I was in particular referring to the dependence on width bing tied to input dimension. I appreciate the clarification, and am willing to increase this.
> >
> > > Moreover, we would appreciate more concrete feedback as to why the reviewer is "not convinced by the technical depth of the main result".
> >
> > The reason for saying this was primarily based on the fact that the lower bound construction essentially uses the flatness of the ReLU allowing it to represent functions that are non-zero only in a small interval where the width of the interval is governed by the Lipschitz constant. Here, the particular NN architecture is not super important (as in depth, width etc.) apart from allowing for a blow up of the Lipschitz constant based on the norm bounds, which could be done by just having a higher norm bound in the first layer itself. Furthermore, as the authors point out in the response to Reviewer VFv9, it is not clear if this extends to smoother/analytical activations like sigmoid.
> >
> > > Our code can readily be executed for any dimension. ... Moreover, we face the severe issue that the uniform accuracy cannot be reliably evaluated in higher dimensions.
> >
> > I agree with the authors on the computational issues with running the experiments. Their comments exactly highlight the issues I had with the usefulness of the experiments. It still is not clear to me what the value of the experiments is wrt the theory.

---

> > > ### Author Response · Authors · 2022-11-22
> > > **Second response to reviewer Jti4**
> > >
> > > We thank you for your response and willingness to increase your score. We are very happy to have addressed several of your concerns. Please find our answers to your remaining questions in the following:
> > >
> > > > **The reason for saying this was primarily based on the fact that the lower bound construction essentially uses the flatness of the ReLU allowing it to represent functions that are non-zero only in a small interval where the width of the interval is governed by the Lipschitz constant. Here, the particular NN architecture is not super important (as in depth, width etc.) apart from allowing for a blow up of the Lipschitz constant based on the norm bounds, which could be done by just having a higher norm bound in the first layer itself. Furthermore, as the authors point out in the response to Reviewer VFv9, it is not clear if this extends to smoother/analytical activations like sigmoid.**
> > >
> > > The behaviour for different activation functions will be an interesting follow-up work (both the theoretical as well as empirical behavior) and, as mentioned in the response to Reviewer VFv9, we conjecture that there might be a different behavior, in particular for analytic activation functions.
> > >
> > > Regarding piecewise linear activation functions (such as the ReLU), you correctly point out that there is some flexibility in designing the counterexample. It was in fact surprising to us that we could achieve such explicit, almost matching bounds, which, in particular, show that our construction is near optimal. Furthermore, a certain width-depth ratio, depending on the regularization constraint and dimension, seems to be necessary to obtain these rates. We also want to point out that the Lipschitz constant does not need to blow up while still yielding intractable rates for the $L^p$-reconstruction error with $p \gg 1$ (depending on $d$). Finally, we believe that our proof techniques (e.g., estimating sample complexities via average case analysis) will be a useful introduction to the community (such techniques have so far mostly been used in the information-based complexity community).
> > >
> > >
> > > > **I agree with the authors on the computational issues with running the experiments. Their comments exactly highlight the issues I had with the usefulness of the experiments. It still is not clear to me what the value of the experiments is wrt the theory.**
> > >
> > > The main contribution of our paper is of theoretical nature. Nevertheless, we were keen to see how the reconstruction error behaves in a more practical setting. Our experiment design choices are governed by the following constraints:
> > >
> > > * We need to replace the (uncountable) target class $U$ by a finite number of teacher networks.
> > > * We also need to restrict ourselves to a finite number of algorithms, where  we chose  different student architectures optimized by variants of gradient descent.
> > > * We can only reliably estimate the uniform norm in low dimensions.
> > > * Finally, the number of choices for the teacher networks and algorithms are restricted by our computational budget.
> > >
> > > Within in these natural constraints and our computational budget, we estimated how the reconstruction errors can behave in practice. Of course, one then cannot expect to recover the precise theoretical rates. As described in our paper, the rates worsen when considering more teacher networks and improve when considering further algorithms. However, we tried to find a reasonable trade-off (based on substantial computational resources) and also did not observe much variance when changing our setting.
> > >
> > > The take-aways of the experiments are as follows:
> > >
> > > 1. In Figure 2, we clearly see the gap between the uniform and the $L^p$ (p=1,2) errors, which is in line with our theory. One can also estimate the rate realized in practice. Similar to previous work by Fiedler et al. (2023) and Adcock and Dexter (2021), one further observes that, even in one dimension, the uniform reconstruction error seems to stagnate at a precision which might be insufficient for applications.
> > >
> > > 2. Astonishingly similar to our proof strategy, we observe in Figure 3 that these errors are mainly caused by spikes of the target function between the samples.
> > >
> > > Another contribution can also been seen in our code (featuring very flexible experiment execution/tracking and hyperparameter searches using Ray Tune), which can be used to compare the rates for other settings. One of such settings, as described above, will the be extension to other activation functions.

---

> > > > ### Comment · Reviewer_Jti4 · 2022-11-28
> > > > **Response**
> > > >
> > > > Thank you for the response, I appreciate the clarifications. While I do understand the choices the authors made on the experimental section, I still do not think it adds much to the theoretical results.
> > > >
> > > > > We also want to point out that the Lipschitz constant does not need to blow up while still yielding intractable rates for the $L^p$-reconstruction error with $p >> 1$ (depending on $d$).
> > > >
> > > > I think the Lipschitz constant in the relevant $q$-norm would blow up in the construction. I'm not sure I follow. Please clarify.

---

> > > > > ### Author Response · Authors · 2022-11-30
> > > > > **Third response to reviewer Jti4**
> > > > >
> > > > > Thank you for your response! Let us answer your remaining question in the following:
> > > > >
> > > > > > **I think the Lipschitz constant in the relevant $q$-norm would blow up in the construction. I'm not sure I follow. Please clarify.**
> > > > >
> > > > > 1. Let us first construct bump functions with a fixed Lipschitz constant, see Lemma A.2:
> > > > >
> > > > >
> > > > >     Let us choose $L,B\in\mathbb{N}\_{\ge 3}$, $c>0$, and $q\ge 2$ such that $\lambda = \sqrt{s} > c^L (B^{1-\frac{2}{q}})^{L-1}/12$.
> > > > >     For instance, we could choose $q=2$ and $c=12^{1/L}$.
> > > > >     Then, Lemma A.2 and a simple computation show that
> > > > >     $h=\nu \frac{\lambda}{Ms} \cdot \vartheta^{(s)}\_{M,y} \in U= \mathcal{H}^{q}\_{(d,B,\cdots,B,1),c}
> > > > >     $ for all choices of $M\in \mathbb{N}$, $s\le \min\left\\{ \frac{B}{3},d\right\\}$, $\nu\in\\{\pm 1\\}$, and $y\in [0,1]^d$ with Lipschitz constant $\operatorname{Lip}_{\ell^2}(h)$ (uniformly) bounded by 1.
> > > > >
> > > > > 2. We now identify scenarios where reconstruction of corresponding networks in $U$ is intractable and where the proof only uses bump functions with Lipschitz constant bounded by 1, see Theorem A.5:
> > > > >
> > > > >
> > > > >     Let us abbreviate $\varepsilon=\operatorname{err}_m^{MC}\big(U;L^p([0,1]^d)\big)$. For the choice of $\lambda$ from above, Theorem A.5 shows that $\varepsilon \ge \frac{\sqrt{s}/2}{(64s)^{1+\frac{s}{p}}} \cdot m^{-\frac{1}{p}-\frac{1}{s}}$. For instance, for $p=\infty$ and $\varepsilon \le \frac{1}{256\sqrt{s}}$, this yields the bound $m\ge (128\varepsilon \sqrt{s})^{-s} \ge 2^{s} $. This constitutes an intractable number of samples already in low dimensions, e.g., $d=s=100$ (for $B\ge 300$), with practically relevant accuracies of $\varepsilon\le \frac{1}{256\sqrt{s}}\approx 0.0004$.
> > > > >
> > > > >
> > > > >     Similarly, one can prove the existence of further intractable scenarios for $p<\infty$ and sensible choices of the other parameters.
> > > > >
> > > > > In summary, this shows that the Lipschitz constant of the bump functions does *not* need to blow up while still requiring an intractable number of samples.

---

> > > > > > ### Comment · Reviewer_Jti4 · 2022-11-30
> > > > > > **Response**
> > > > > >
> > > > > > Thanks for the clarification! I understand that the Lipschitzness in the $\ell_q$-norm can be bounded while paying an exponential dependence on dimension $d$ for $p= \infty$ recovery. The exponential in $d$ comes from the fact that getting a sample in the small ball has exponentially low probability. In your example, the parameters are set such that there is no $L$ dependence anymore in the bound in order to ensure the Lipschitzness is bounded by a constant. This exponential dependence was what I was referring to.

---

> > > > > > > ### Author Response · Authors · 2022-12-02
> > > > > > > **Fourth response to Reviewer Jti4**
> > > > > > >
> > > > > > > Thank you for your response! Let us answer your remaining question in the following:
> > > > > > >
> > > > > > > > **In your example, the parameters are set such that there is no $L$ dependence anymore in the bound in order to ensure the Lipschitzness is bounded by a constant. This exponential dependence was what I was referring to.**
> > > > > > >
> > > > > > > The exponential dependence in $L$ actually requires a Lipschitz constant that also scales exponentially in $L$. This is however not due to a limitation in our construction but rather a fundamental property that cannot be avoided: As implied by Lemma B.3 in our paper, if all the functions of interest had a Lipschitz constant bounded by $C(d,L)$, then Lemma B.3 shows that an algorithm based on $m$ point samples can achieve $L^\infty$ recovery error $2 \sqrt{d}  C(d,L) \cdot m^{-1/d}$ over the entire class.
> > > > > > >
> > > > > > > Hence, for proving that the optimal reconstruction error is lower bounded by $C(d,L) \cdot m^{-1/d}$, one needs to exhibit functions in the considered class which have Lipschitz constant at least $\frac{C(d,L)}{2 \sqrt{d}}$.
> > > > > > > In other words, it is not possible to construct counterexamples with bounded Lipschitz constant such that the optimal error explodes with $L$.

---

> > > > > > > > ### Comment · Reviewer_Jti4 · 2022-12-02
> > > > > > > > **Final Response**
> > > > > > > >
> > > > > > > > Yes, I agree with what you said. This was my original comment, that the dependence on $L$ comes only from scaling this Lipschitz constant based on the chosen norm constraints which could be avoided by a different set of norm constraints on the weights. So I see the main takeaway as the scaling with $d$.
> > > > > > > >
> > > > > > > > I will maintain my score (I increased it from 3 to 5 previously based on our discussions). I still feel the setting is restrictive given the $\ell_\infty$ guarantee and the experiments are not super insightful. Having said that, I still think the paper has interesting ideas, and would not be disappointed to see the paper being accepted.

---

> > > > > > > > > ### Author Response · Authors · 2022-12-02
> > > > > > > > > **Thank you**
> > > > > > > > >
> > > > > > > > > Thank you for your response. We appreciate your comments and your engagement.

---

> ### Author Response · Authors · 2022-11-19
> **Response to reviewer Jti4 (2/2)**
>
> >
> > **What are the differences with the constructions in Telgarsky 2015? I understand that the work deals with purely approximation and not sample complexity, but they do bound the total error in the interval.**
>
> First, Telgarsky only considers the *classification* error with respect to the width and depth for a specific n-alternating-point problem in $d=1$. In contrast, we consider a *regression* problem with adaptive sampling in arbitrary dimension constrained by corresponding regularization on the network parameters.
>
> Second, the proof strategy of Telgarsky is to construct sawtooth functions by composing hat functions, whereas we construct localized and scaled hat functions that are as large as possible, subject to a certain prescribed regularization (i.e., size constraint on the network weights). Most importantly, Telgarsky studies how the architecture affects the expressiveness in terms of the number of linear pieces, whereas we are interested in how the architecture/regularization/norm affects the learnability of a class of neural networks based on point samples.
>
> >
> > **Overall, in my opinion, the worst-case nature of the result makes it not very interesting. The requirement of $\ell^\infty$-bounded error becomes extremely strong, and not very insightful for realistic models. As the authors point out in the limitations, studying this in a more reasonable sub-class of neural networks or understanding uniform error restricted to some manifold or under some perturbation model would be more useful and interesting.**
>
> We definitely agree that these are very interesting further direction. However, we think it is important to have provided a rigorous starting point. As can be seen from our proofs the corresponding *quantitative* results already seem to require some careful analysis. Our results also hold for the whole range $p\in[1,\infty]$ and, in particular, show how fast the problem becomes intractable (depending on the architecture and regularization) when $p\to \infty$.
>
> >
> > **Correctness: 3: Some of the paper’s claims have minor issues. A few statements are not well-supported, or require small changes to be made correct.**
>
> We would be curious to know which of our claims you think have minor issues such that we could fix those.

---

### Official Review · Reviewer_81vF · 2022-10-31

**Confidence:** 2
**Clarity, Quality, Novelty And Reproducibility:** 1. I suggest formally giving the defi…
**Correctness:** 4
**Technical Novelty And Significance:** 3
**Empirical Novelty And Significance:** 3
**Recommendation:** 6

**Details Of Ethics Concerns:**

No.

**Strength And Weaknesses:**

Strength:  This paper study the sample complexity of learning high uniform accuracy neural networks. Uniform accuracy is not a common criterion that people analyze in statistical learning theory, but it can help to understand the stability of the network, which I believe is an important contribution to the deep learning community.


Weakness:  While I’m not familiar with uniform accuracy-related work, I feel some of the important lines of literature are missing. For example, I’m wondering whether the author can first provide some detailed literature results regarding the difference between sample complexity of average accuracy vs uniform accuracy on simpler hypothesis classes, such as linear classifier, polynomial classifier, etc? Moreover, I believe how neural network approximate function is another line of related work related to this paper, which should be discussed and compared with the results.


**Summary Of The Paper:**

This paper provides sample complexity upper bound and lower bound for learning high uniform accuracy ReLU networks. It shows that any learning algorithm recovers neural networks to achieve high uniform accuracy needs intractably many samples (exponentially depending on the input dimension, network width and network depth).


**Summary Of The Review:**

Overall, I believe this is an interesting paper with good contribution that considers the sample complexity of neural networks for uniform accuracy.

---

> ### Author Response · Authors · 2022-11-19
> **Response to reviewer 81vF (1/2)**
>
> We thank you for your review and the positive feedback. Please find our answers to your suggestions and questions below.
>
> >
> > **While I’m not familiar with uniform accuracy-related work, I feel some of the important lines of literature are missing. For example, I’m wondering whether the author can first provide some detailed literature results regarding the difference between sample complexity of average accuracy vs uniform accuracy on simpler hypothesis classes, such as linear classifier, polynomial classifier, etc?**
>
> We agree that a discussion of learning with uniform accuracy for other hypothesis classes is important context for our results.
> In fact, we discuss this in Remark 1.3, where we state that uniform learning is possible for linear functions and more generally for (sparse) polynomials, and also for certain RKHS; we provide literature references for each. Furthermore, we note that if the hypothesis class forms (or is contained in) a vector space of dimension $D$, then it follows by standard linear algebra arguments that one can recover each element of the hypothesis class *exactly* from $D$ noise-free measurements. Hence, in this case there is no difference between uniform accuracy and $L^2$-accuracy. This in particular applies for linear regression.
>
>
> >
> > **Moreover, I believe how neural network approximate function is another line of related work related to this paper, which should be discussed and compared with the results.**
>
> We are well familiar with large parts of the literature studying the approximation properties of neural networks. However, we don't see an immediate connection to the results in the present paper. What is needed for (the proofs) in this paper is that ReLU networks can implement certain very special functions which we then use to prove our hardness results. Since we do not see a close connection between our paper and the approximation theory of neural networks (and for reasons of space), we refrained from discussing this literature.
>
> >
> > **I suggest formally giving the definition of uniform accuracy instead of burying it in the text.**
>
> While our results in the first Section are purposefully written in a more informal but accessible way, we always refer to the detailed definitions and theorems in Section 2. However, we appreciate your feedback and added a corresponding definition on page 2.
>
> >
> > **A detailed comparison between the sample complexity of NN under uniform accuracy and under normal average accuracy proposed in the literature, is necessary**
>
> Thanks for this suggestion. We added a brief remark regarding this in Section 1.1. Further comments in this direction can be found in Remark 2.3.
>
> >
> > **The proof sketch in the main paper can be clearer, i.e. first give the definition of bump functions. Some of the math notation in the proof sketch is not introduced.**
>
> We would definitely like to add more details, however we have been constrained by the page limit. We believe that the *bump function* $\vartheta_{M,y}$ is the only notation which is not properly introduced. While we reference the corresponding Lemma A.1, we also define the function by its properties, namely its support and its $L^p$-norm. Note that these are also the only properties which are needed in the proof.
>
> >
> > **What are the x-axis and y-axis of figure 3, also the discussion is not very clear to me.**
>
> This figure does not depict errors, but directly depicts the target function that should be learned (based on point samples) and the model that is learned based on these point samples. The x-axis and y-axis in this figure therefore really represent x-values (possible inputs, from the real numbers since this figure is for $d=1$) and y-values (outputs of the target function and the learned model for the given inputs). We hope this clears up the confusion and also makes the discussion (i.e., the caption of the figure) more clear.

---

> ### Author Response · Authors · 2022-11-19
> **Response to reviewer 81vF (2/2)**
>
>
> >
> > **Though the theorem suggests sample complexity exponentially depends on the dimension, I'm wondering whether the experiments can be more flexible. Only consider dimension=1 and 3 looks too simple and hardly gives any insight for real-world application.**
>
> Our code can readily be executed for any dimension. However, we already executed 8640 experiments for these plots and are unfortunately restricted by our computational resources. Moreover, we face the severe issue that *the uniform accuracy cannot be reliably evaluated in higher dimensions*. To sample every $\epsilon$-neighborhood in $d$ dimensions, one would need $\epsilon^{-d}$ many samples which is already infeasible for small dimension. For our experiments, we already use $2^{24}$ samples to obtain a good estimate for dimensions $3$ and $5$.
>
> >
> > **I'm not following the motivation that connects uniform accuracy with adversarial robustness, in the sense that this paper considers a teacher-student setting, so all the samples are guaranteed to generate from the same neural network, whereas for adversarial robustness, even considering the standard Lp perturbation attack, there's no guarantee that the adversarial examples and the clean samples all come from the same network.**
>
> We agree to some extent with the assessment regarding the direct application of our results to adversarial robustness and reformulated the text in the discussion section accordingly. Our results show, roughly speaking, that knowledge of labels (even noise-free ones) on a number of samples that does not scale exponentially on $L$ and $d$ does not carry sufficient information to determine a neural network up to small uniform error.

---

### Official Review · Reviewer_VFv9 · 2022-11-02

**Confidence:** 3
**Correctness:** 4
**Technical Novelty And Significance:** 4
**Empirical Novelty And Significance:** 2
**Recommendation:** 8

**Clarity, Quality, Novelty And Reproducibility:**

See the above section.

Besides, I also have a series of questions:

1. In Theorem 1.1, it is written ``$L$ layers with width up to $3d$''. Is the statement wrong (maybe the correct statement would be $L$ layers with width at least $3d$)? I think increasing the width should only make the network function more complicated.
2. In the main theorem, when $q=2$, it seems that $\Omega$ will have different values under the two different calculations in equation (3). Are the constants $\frac 1 {4\times 3^{2/q}}$ and $\frac 1 {24}$ correct? What is the reason for this inconsistency?
3. Theorem 2.2 requires the assumption that $L\ge 3$. I wonder why it cannot be applied for two-layer networks. Are two-layer networks fundamentally different from networks with $L\ge 3$ layers in proving this theorem?
4. The authors said that Theorem 2.2 only applies for ReLU neural networks. I am quite interested in the assumption and would like to know more about it: what are the difficulties in generalizing the result to other activation functions? Do you think/conjecture that Theorem 2.2 can still apply in general settings?
5. Regarding Figure 3: it is said that this paper focuses on ReLU networks. However, the blue curve in Figure 3 is smooth, which is strange to me. Can the authors provide an explanation?
6. The authors use simple vector $\ell_q$ norm to constrain the model's parameters. While it is certainly fine (not a weakness), I may wonder if it is possible to use bounded matrix norm per layer to constrain the model's parameters since it is finer than the coarse vector norm and takes into consideration the layered network structure.

Miscellaneous minor issues:
The first line in page 8: what is equation (2.1) linked to?

**Strength And Weaknesses:**

**Strength**:

I feel the presented theoretical result is interesting. The result gives us a better understanding of the complicated nature of the neural network function class which is fundamentally different from classic function classes, such as linear models or polynomials. It shows that the representation of a neural network cannot be inferred from a tractable number of input-output samples even if the number of network parameters is small. I partially agree with the authors that the result may be useful for future works in understanding several important topics, e.g., why modern neural networks are vulnerable to adversarial examples. I also agree that learning functions to minimize the uniform error $L^\infty$ instead of $L^2$ is *necessary* for some applications such as solving partial differential equations. So in this respect, I think the present theoretical result is significant.

This paper is well-written and clear. Although the theoretical bound involves many problem-dependent parameters in a complicated manner, it is well-explained using carefully presented remarks. Moreover, the simplified version (Theorem 1.1) gives readers an overall picture of the more advanced result, while being easy to understand. The authors also present a proof sketch that is helpful in gaining insights into why the result holds. The discussions in this paper are particularly comprehensive.

**Weaknesses**:
1. I think there is a mismatch between the setting in this paper and the commonly used settings in learning real-world function classes. In practice, it is unlikely that the target function is represented by a neural network similar to the counterexample in this paper. If I understand correctly, the reason why the sample complexity must scale like $\Omega(\epsilon^{-d})$ is that neural networks can represent ''bump functions'' which are highly local. Therefore, if there are no samples in the ``bump'' area, the target function clearly cannot be learned. However, such non-smoothness may hardly appear in practice. Therefore, the current theory may not shed light on practical settings. I think it is an important and natural assumption that the target function to be learned is ''smooth'' (rather than all functions that can be represented by neural networks).\
 Although the authors said that ''this is satisfied in several applications of interest, e.g., model extraction attacks (Tramèr et al., 2016; He et al., 2022) and teacher-student settings (Mirzadeh et al., 2020; Xie et al., 2020)'', the target functions there are *well-trained* neural networks rather than any possible one in the neural network function class. Therefore, these cases may not be the same as the one studied in this paper.

2. The authors also said that the result is related to the adversarial vulnerability of neural networks. While I partially agree with the potential implications of the result in this paper, I think the current setting is still quite different from the setting of adversarial examples. In particular, Theorem 1.1 may only show that neural networks can represent highly non-smooth functions with many spikes, for which adversarial examples exist. However, given finite samples, why are neural networks prone to learn bump/spike functions rather than smooth functions? It is because of the inductive bias or training algorithms? I think currently it is still not clear how the result in this paper relates to the adversarial vulnerability of neural networks.

3. While I am generally satisfied with the case when $q=\infty$ (i.e., the setting of bounded weight coefficients), I think the current bound for $q<2$ may be vacuous and unreasonable. In particular, in this case $\Omega$ scales like $d^{-d}$, which means that when fixing $\epsilon$, the bound $\left(\frac {\Omega}{64d\epsilon}\right)^d$ vanishes as $d\to \infty$. Namely, the lower bound becomes trivial when $d$ is large. This is clearly impossible since the problem becomes harder when $d$ is larger. Therefore, the bound for the case of $q<2$ is very loose and may not really make sense. Note that there is a large gap between the lower bound and the upper bound for $q\le 2$ (Theorem 2.4, which does not vanish when $d$ is large). In another way, the main theorem only applies when $\epsilon=o(1/d)$. For the high dimensional setting which usually occurs in practice, the bound becomes vacuous.

**Summary Of The Paper:**

This paper presents a fundamental impossibility result associated with the neural network function class. It is proven that given a specific network architecture (e.g., depth and width) with ReLU activation function and norm-bounded parameters, there exists a neural network function u such that u cannot be determined *efficiently* from (possibly adaptive) input-output sample pairs, where the needed sample size to determine u can be lower bounded by a term that scales exponential in the input dimension d. This result is equivalent to the following one: given a specific network architecture with ReLU activation function and norm-bounded parameters, to learn a target function uniformly well with error $\epsilon$ (where the error is measured to be the $L^\infty$ distance between the learned function and the target), the sample complexity must scale like $\Omega(\epsilon^{-d})$.

The paper discussed a quite general setting with the error defined to be $L^p$ distance for any $p\in [1,\infty]$. The above exponential bound happens exactly when $p=\infty$, which contrasts the common squared-error setting $p=2$, for which sample-efficient learning is possible.

Moreover, the dependency of several problem-dependent parameters is made explicit in the bound, such as the network layer $L$, the hidden dimension $B$, and the scale of the parameters $c$ under $\ell_q$-norm. The authors gave a fruitful discussion of their results and also presented an upper bound. They finally conducted experiments to verify the theoretical results.

**Summary Of The Review:**

Overall, I think this paper is interesting. The result is well-presented, and the writing is clear. Nevertheless, there are several weaknesses raised above. I hope the authors can clarify them and I will reevaluate this paper based on authors' response.

---

> ### Author Response · Authors · 2022-11-19
> **Response to reviewer VFv9 (1/3)**
>
> We are very grateful for your extensive review and the positive feedback. We will comment on some of the weaknesses and answer your questions in the following:
>
> >
> > **I think it is an important and natural assumption that the target function to be learned is ''smooth'' (rather than all functions that can be represented by neural networks).
> Although the authors said that ''this is satisfied in several applications of interest, e.g., model extraction attacks (Tramèr et al., 2016; He et al., 2022) and teacher-student settings (Mirzadeh et al., 2020; Xie et al., 2020)'', the target functions there are well-trained neural networks rather than any possible one in the neural network function class. Therefore, these cases may not be the same as the one studied in this paper.**
>
> We agree that these sharp bumpy functions appear unnatural at first. However, it is hard to mathematically quantify the regularity of real-world target functions. In fact, it is difficult to fully exclude such pathological bump functions: our proofs reveal that neural networks do not need large parameter magnitudes to represent them. Moreover, our proof does not need these bump functions themselves, but rather a sum of a fixed function and such bump functions.
> For instance, in classification problems one could at least expect steep classification boundaries. Also, *in scientific applications, one often encounters functions with high frequency content*. Figures 1 and 3 in our paper and the works by Fiedler et al. (2023) and Adcock and Dexter (2021) cited in our paper provide evidence that a well-trained (teacher) network can indeed exhibit such bump functions. We added a remark stating this in the revised version.
> We finally note that in fact the bump functions need not be necessarily very sharp. In fact, as can be seen in Lemma A.2, the Lipschitz constant of the bump functions can be uniformly bounded but the exponential (in $d$) dependence of the sample complexity still holds true. This in particular means that regularizing the Lipschitz constant will not really help.
> If the target function is (assumed to be) smooth in the classical sense (i.e., $C^k$), it is known that polynomials already provide an optimal approximation rate. Therefore, in this setting it would seem that polynomials would be a more natural class than neural networks. However, it is well known that the class of smooth functions cannot be approximated without the curse of dimension which is why (classical) smoothness is generally not a very useful prior in high dimensions.
>
> >
> > **In particular, Theorem 1.1 may only show that neural networks can represent highly non-smooth functions with many spikes, for which adversarial examples exist. However, given finite samples, why are neural networks prone to learn bump/spike functions rather than smooth functions? It is because of the inductive bias or training algorithms? I think currently it is still not clear how the result in this paper relates to the adversarial vulnerability of neural networks.**
>
> We agree to some extent with the assessment regarding the direct application of our results to adversarial vulnerabilities and reformulated the text in the discussion section accordingly. As outlined above, it has been *empirically* observed that such bump functions can (and do) in fact occur when training neural networks. We are conducting further research in this direction, but at this point it seems unclear *why* such phenomena occur.
> On the other hand, one can prove that *for sufficiently large width* gradient descent leads to smooth, spline-like interpolation for two-layer networks, see the results of [Shevchenko et al. (2021)](https://arxiv.org/abs/2111.02278).

---

> ### Author Response · Authors · 2022-11-19
> **Response to reviewer VFv9 (2/3)**
>
> >
> > **[...] I think the current bound for $q<2$ may be vacuous and unreasonable. In particular, in this case $\Omega$ scales like $d^d$, which means that when fixing $\epsilon$, the bound $\left(\frac{\Omega}{64d\epsilon}\right)^d$ vanishes as $d\to\infty$. Namely, the lower bound becomes trivial when $d$ is large. This is clearly impossible since the problem becomes harder when $d$ is larger. Therefore, the bound for the case of $q<2$ is very loose and may not really make sense. Note that there is a large gap between the lower bound and the upper bound for $q\le 2$ (Theorem 2.4, which does not vanish when $d$ is large). In another way, the main theorem only applies when $\varepsilon = o(1/d)$ . For the high dimensional setting which usually occurs in practice, the bound becomes vacuous.**
>
> We mostly agree with your assessment; our thoughts on this point are elaborated in Remark B.4 in the appendix (which is referenced after Theorem 1.4 in the main text). We would like to add two points here: First, the more precise version of Theorem 1.1 given in Theorem 2.2 shows for $q \leq 2$ that one gets a lower bound of $m \geq \left(\frac{c^L}{\kappa \cdot s^{2/q} \cdot \epsilon}\right)^s$ with an absolute constant $\kappa > 0$, for any $s \leq \min (d, B/3)$, where $B$ is the width of the considered networks. This shows that the bound indeed gets worse for large $d$ (since one has more flexibility in choosing $s$). Second, despite this observation, there is indeed a gap between our upper and lower bounds in this setting; we conjecture that neither the lower- nor the upper bound are fully sharp, but we have been unable to improve them to the point of being fully sharp.
>
> >
> > **In Theorem 1.1, it is written “$L$ layers with width up to $3d$”. Is the statement wrong (maybe the correct statement would be layers with width at least $3d$)? I think increasing the width should only make the network function more complicated.**
>
> This is correct as stated. The point of the theorem statement is that even though the considered network class is “not too complicated” (since we allow a width of at most $3d$), we still get the stated lower bound for the required number of samples.
> We also want to emphasize that our theorem considers the set of *all* neural networks with width up to $3d$ in each of the hidden layers.
>
> >
> > **In the main theorem, when $q=2$, it seems that will have different values under the two different calculations in equation (3). Are the constants $\frac{1}{4 \times 3^{2/q}}$ and $\frac{1}{24}$ correct? What is the reason for this inconsistency?**
>
> Thank you for the precise inspection. This is a proof artifact and we were not able to construct a neural network with a matching bound. However, this constant factor is inconsequential for our results. In fact, we already briefly mentioned this discrepancy before Lemma A.3 in the appendix.
>
> >
> > **Theorem 2.2 requires the assumption that $L\ge 3$. I wonder why it cannot be applied for two-layer networks. Are two-layer networks fundamentally different from networks with layers in proving this theorem?**
>
> Our construction of the bump function requires three layers, see Appendix A.1. One can prove that no (non-trivial) two-layer ReLU network can represent a compactly supported function, see Proposition 4.1 in [http://pc-petersen.eu/Neural_Network_Theory.pdf](http://pc-petersen.eu/Neural_Network_Theory.pdf), which is based on a result by Blum and Li (1991). We added a corresponding note in the revised version before Lemma A.2 in the appendix. While one could maybe modify the proof to still work for two-layer networks, we decided to focus on the more practically relevant case of networks with at least two hidden layers.

---

> ### Author Response · Authors · 2022-11-19
> **Response to reviewer VFv9 (3/3)**
>
> >**The authors said that Theorem 2.2 only applies for ReLU neural networks. I am quite interested in the assumption and would like to know more about it: what are the difficulties in generalizing the result to other activation functions? Do you think/conjecture that Theorem 2.2 can still apply in general settings?**
>
> It is straightforward to extend the results to continuous piecewise linear activation functions with at least one breakpoint, see, e.g., Proposition 1 by [Yarotsky (2017)](https://arxiv.org/pdf/1610.01145.pdf). However, this would make the results more technical without yielding additional insight. We conjecture that similar results to ours can be shown for piecewise polynomial activation functions. That said, we conjecture that the case of certain smooth (and in particular, analytic) activation functions might show a different behavior, in the sense that exact recovery might be possible once the number of sampling points exceeds (a constant multiple of) the number of parameters of the neural network; we are currently investigating this. One important property of networks with analytic activation function is that they are never compactly supported (unless they vanish everywhere); because of this, the constructions that we employ to prove our hardness results cannot be applied for such activation functions.
>
> >**Regarding Figure 3: it is said that this paper focuses on ReLU networks. However, the blue curve in Figure 3 is smooth, which is strange to me. Can the authors provide an explanation?**
>
> It indeed *looks* smooth and, back then, we also double-checked the settings. The reason is that the best performing student network, out of widths $\\{32, 512, 2048\\}$, for this setting had width $2048$ (as compared to the teacher with width $32$). The large number of parameters renders the plot "almost" smooth, when not zooming in even further.
>
> >**The authors use simple vector norm to constrain the model's parameters. While it is certainly fine (not a weakness), I may wonder if it is possible to use bounded matrix norm per layer to constrain the model's parameters since it is finer than the coarse vector norm and takes into consideration the layered network structure.**
>
> This would certainly be possible; we mainly consider the $\ell^q$-norm of the entries for simplicity and since regularizing using the $\ell^2$-norm of the network weights is a common regularization strategy. A further point is that once one starts considering general matrix norms one could get even more variables, since one could consider norms of the form $\|A\|\_{\ell^{q_1} \to \ell^{q_2}}$. Our upper bounds will (at least in spirit) apply whenever one has bounds on $\|A\|\_{\ell^q \to \ell^q}$ for some $q$; see (the proofs of) Lemmas B.2 and B.3. Furthermore, our lower bound for $q = 2$ provides a hardness result for the recovery using point samples of networks with $\ell^2$ (i.e., Hilbert-Schmidt) control over the network weights on each layer. This is a more stringent assumption than control over the $\ell^2 \to \ell^2$ matrix norms of the weights, and hence the same hardness result also holds for networks with this matrix norm being bounded. We don't think that the resulting lower bound can be significantly improved.
>
> >**Miscellaneous minor issues: The first line in page 8: what is equation (2.1) linked to?**
>
> Thank you for spotting this typo! We fixed it in the revised version.

---

> ### Comment · Reviewer_VFv9 · 2022-11-28
> **Thank you for your detailed response.**
>
> I would like to thank the authors for their detailed response. I think several of my concerns have been addressed. In particular, it is quite interesting to see that exact recovery might be possible for smooth activations once the number of sampling points exceeds (a constant multiple of) the number of parameters of the neural network.
>
> Overall, I think **a score of 7 is appropriate**. However, since ICLR does not have such a choice, I may leave the current score unchanged.

---

### Official Review · Reviewer_8e6q · 2022-11-04

**Confidence:** 3
**Clarity, Quality, Novelty And Reproducibility:** 1. Clarity
**Correctness:** 4
**Technical Novelty And Significance:** 3
**Empirical Novelty And Significance:** Not applicable
**Recommendation:** 6

**Strength And Weaknesses:**

Strengths

1. The results are quantitative where both the upper and lower bound are non-asymptotic. The proof is clean and constructive.
The paper is very clearly written. Special care was given to compare to existing work and to discuss specific results in order to avoid over-claiming.
2. Assumptions made are natural and rather weak. The learning problem is also natural and relevant (teacher-student learning).
Relevant empirical study to substantiate theoretical claims.

Weaknesses

1. The results are not very surprising as learning in L-infinity is known to be hard.
2. Some discussions can be useful regarding the upper bound result (of Theorem 2.4).
3. The results are only truly novel in the L-infinity setting, which is a restrictive setting (mainly for security, safety-critical applications), as learning is known to be tractable in L-p where p is finite. (This is acknowledged by the authors.)
4. The constructed hard function for the lower bound proof makes use of sharp bumpy hat functions, which is not natural in real-world settings.
5. I would really appreciate a more in depth comparison to SQ learnability literature since they share the common setting of studying sample complexity (as opposed to runtime complexity under fixed sample size). The authors, however, did a good job comparing their results in the PAC learnability setting.

**Summary Of The Paper:**

The paper provides a theoretical study on the difficulty of learning an unknown ReLU network with a fixed number of layers by another ReLU network with the same number of layers. While previous attempts usually look at NP-hardness of the optimization process of this task, even allowing for known data distribution, the paper tackles the problem from a learnability/information theoretic point of view and studies bounds on sample complexity that any learning algorithm can achieve. The end result is a separation between L-infinity learnability and Lp learnability when p is finite. In particular, when requiring that the learned network is close in L-infinity norm, the sample complexity scales at least exponentially in depth and input dimension, even if one allows for arbitrary runtime in the learning algorithm.








**Summary Of The Review:**

Despite the limited setting under which the main results are proven (learning under L-infinity) and slightly unrealistic counterexample for the lower bound (which is rather common in the literature, in fact), the paper is very clearly written (even in the proof), gives concrete, quantitative, and comprehensive results under mild assumptions and is, in general, an interesting and well-conceived theoretical paper. Therefore, I recommend acceptance.

---

> ### Author Response · Authors · 2022-11-19
> **Response to reviewer 8e6q**
>
> We thank you for your detailed and positive review and will comment on some of the weaknesses below:
>
> >
> > **The results are not very surprising as learning in L-infinity is known to be hard.**
>
> It holds that $\\|f-g\\|\_{L^p(\mathbb{P})} \le \\|f-g\\|\_{L^q (\mathbb{P}) }$ for $1\le p\le q\le \infty$, measurable functions $f$ and $g$, and any probability measure $\mathbb{P}$. In this sense, learning naturally gets harder for increasing $p$. However, we mention in Remark 1.3 that for other hypothesis classes (which are also universal), such as polynomials or certain RKHS, uniform reconstruction can nevertheless be tractable. As also acknowledged in your review, our work is the first to quantify the opposing behavior for the class of neural networks. Our results also quantify how quickly the problem becomes intractable (depending on the architecture and regularization) when $p\to \infty$.
>
> >
> > **Some discussions can be useful regarding the upper bound result (of Theorem 2.4).**
>
> We already had some discussion of the results and the gap between the lower and upper bound in the appendix, see Appendix B and, in particular, Remark B.4 (which is referenced after Theorem 1.4). However, we agree that the main text should at least convey the main idea of the proof and we added this to the revised version.
>
> > **The constructed hard function for the lower bound proof makes use of sharp bumpy hat functions, which is not natural in real-world settings.**
>
> We agree that these sharp bumpy functions appear unnatural at first. However, it is hard to mathematically quantify the regularity of real-world target functions. In fact, it is difficult to fully exclude such pathological bump functions: our proofs reveal that neural networks do not need large parameter magnitudes to represent them. Moreover, our proof does not need these bump functions themselves, but rather a sum of a fixed function and such bump functions.
> For instance, in classification problems one could at least expect steep classification boundaries. Also, *in scientific applications, one often encounters functions with high frequency content*. Figures 1 and 3 in our paper and the works by Fiedler et al. (2023) and Adcock and Dexter (2021) cited in our paper provide evidence that a well-trained (teacher) network can indeed exhibit such bump functions. We added a remark stating this in the revised version.
> As you point out in your summary, this type of functions is commonly used in the literature to prove such results.
> We finally note that in fact the bump functions need not be necessarily very sharp. In fact, as can be seen in Lemma A.2, the Lipschitz constant of the bump functions can be uniformly bounded but the exponential (in $d$) dependence of the sample complexity still holds true. This in particular means that regularizing the Lipschitz constant will not really help.
>
> >**I would really appreciate a more in depth comparison to SQ learnability literature since they share the common setting of studying sample complexity (as opposed to runtime complexity under fixed sample size). The authors, however, did a good job comparing their results in the PAC learnability setting.**
>
> A comparison to the SQ (statistical query) literature is provided in the second paragraph of Section 1.1. We have expanded this discussion slightly to include an explicit formulation of a typical result in this area. For us, the main point is still that many algorithms (including SGD) do not strictly fall into the SQ setting (whereas they are covered by our setting) and that the results on sample complexity (with actual point samples, called "label queries" in this community) typically rely on unproven (and difficult) conjectures like the decisional Diffie-Hellman assumption.

---

### Decision · Program_Chairs · 2023-01-20

**Decision:**

Accept: poster

**Justification For Why Not Higher Score:**

- It may not be very surprising that learning in $L^\infty$ norm is harder than learning in $L^2$ norm.
- Since this paper is mostly a hardness result (lower bound result) and seems too theoretical to the broader audience of ICLR, I didn't recommend oral/spotlight presentation for this paper.

**Justification For Why Not Lower Score:**

- The results are important. This paper shows a separation between $L^\infty$ learnability and $L^p$ learnability when $p$ is finite.
- The paper and the proofs are well presented.

**Metareview: Summary, Strengths And Weaknesses:**

Summary
---
This paper studies the difficulty of learning an unknown ReLU neural network with a fixed number of layers by another ReLU neural network with the same number of layers. The authors take a learnability/information theoretic approach and derive bounds on the sample complexity that any learning algorithm must achieve. They show that when the learned network is required to be close in $L^\infty$ norm, the sample complexity scales at least exponentially in the depth and input dimension, even if the learning algorithm is allowed to have arbitrary runtime. In contrast, sample-efficient learning is possible when the error is measured using the $L^2$ distance. The authors also provide upper bounds on the sample complexity and conduct experiments to verify their theoretical results.

Strength
---
- The paper is well-written. The results are non-asymptotic with a clean and constructive proof and are significant for understanding the complicated nature of the neural network function class.
- One of the important insights of the work is that while there exist functions that can be exactly represented by small neural networks, these representations cannot be inferred from samples because of the network's expressiveness. The paper explicitly shows that neural network models with bounded norm coefficients contain a large class of “bump functions” with estimable $L^p$ norms and disjoint supports.

Weaknesses
---
- The constructed hard function for the lower bound proof makes use of sharp bumpy hat functions, which are not natural in real-world settings
- The bound for the case of $q<2$ may be vacuous and unreasonable. There is a large gap between the lower bound and the upper bound for $q\le 2$, and the main theorem only applies when $\epsilon=o(1/d)$ , making it vacuous in the high dimensional setting. The authors mostly agreed with this point in their response.
- It may not be very suprising that learning in $L^\infty$ norm is harder than learning in $L^2$ norm.



**Note From Pc:**

if the above contains the word "oral" or "spotlight" please see: "oral" presentation means -> notable-top-5% and "spotlight" means -> notable-top-25%. As stated in our emails, we are disassociating presentation type from AC recommendations